# Enhanced glacial lake activity threatens numerous communities and infrastructure in the Third Pole

Taigang Zhang [1,2,3], Weicai Wang [1] ✉, Baosheng An[1,4] & Lele Wei [1,2,3]

Glacial lake outburst floods (GLOFs) are among the most severe cryospheric hazards in the Third Pole, encompassing the Tibetan Plateau and surrounding Himalayas, Hindu Kush, and Tianshan Mountains. Recent studies on glacial lake changes and GLOF characteristics and risks in this region have shown scattered and insufficiently detailed features. Here, we conduct an appraisal of the GLOF risks by combining high-resolution satellite images, case-by-case high-precision GLOF modeling, and detailed downstream exposure data. The glacial lake changes from 2018 to 2022 in the region were primarily driven by the accelerated expansion of proglacial lakes. The GLOF frequency has exhibited a significant increasing trend since 1980, with intensified activity in Southeastern Tibet and the China-Nepal border area over the past decade. Approximately 6,353 $km^2$ of land could be at risk from potential GLOFs, posing threats to 55,808 buildings, 105 hydropower projects, 194 $km^2$ of farmland, 5,005 km of roads, and 4,038 bridges. This study directly responds to the need for local disaster prevention and mitigation strategies, highlighting the urgent requirement of reducing GLOF threats in the Third Pole and the importance of regional cooperation.

The Third Pole of the Earth, where includes the Tibetan Plateau and surrounding Himalayas, Hindu Kush, and Tianshan Mountains, is the world's most important water tower[1,2], Due to climate warming, this region has experienced widespread downwasting and the retreat of over 10,000 glaciers over the past three decades[3–5]. Consequently, the increased melting and exposed depressions have facilitated the development of glacial lakes[6]. From 1990 to 2018, a significant number of new and rapidly expanding glacial lakes have been observed in the Third Pole, with a relative area increase of 15.2%[7]. Although glacial lakes capture a small portion of the meltwater contributing to sea level rise and have the potential for hydroelectric power development[8,9], more research interest has been devoted to the appraisal of dangerous glacial lakes and the analysis of destructive glacial lake outburst floods (GLOFs). When a glacial lake is impacted by external forces, such as snow/ice avalanche, landslide, rockfall, etc., or is destabilized by

continuous melting of the underlying buried ice in the moraine dam, it can suddenly release a large volume of water. The resulting flood is rapid and forceful, causes intense erosion along the channel, and is devastating to infrastructure in the GLOF's path[10,11]. A typical Gongbatongshaco GLOF in the eastern Himalayas was documented in 2016. It had a 40-km runout distance and destroyed a hydropower plant in Nepal[12]. Since 1900, more than 110 GLOFs originating from such moraine-dammed lakes have been recorded in the Third Pole and have claimed ~7000 lives[13,14]. The Himalayas, Nyainqentanglha, and Tianshan Mountains serve as the main sources of GLOFs in the Third Pole.

Understanding the historical features of disasters and evaluating their trends are crucial for local disaster response and sustainable development[15]. However, the current knowledge of glacial lake changes and GLOF activities and risks in the Third Pole is still under development. For instance, several recently established independent

[1]State Key Laboratory of Tibetan Plateau Earth System, Environment and Resources (TPESER), Institute of Tibetan Plateau Research, Chinese Academy of Sciences, 100101 Beijing, China. [2]College of Earth and Environmental Sciences, Lanzhou University, Lanzhou 730000, China. [3]Center for the Pan-Third Pole Environment, Lanzhou University, Lanzhou 730000, China. [4]School of Science, Tibet University, Lhasa 850011, China. ✉e-mail: weicaiwang@itpcas.ac.cn

inventories of glacial lakes in the Third Pole have yielded various quantities due to the different glacial lake definitions used[7,16]. The number of glacial lakes reported ranges from 10,000 to 30,000 during 2015–2020. This significant discrepancy hampers our ability to analyze the in-depth change signals and makes the effective assessment of glacial lake hazards and risks challenging. Moreover, the incomplete GLOF dataset makes it challenging to accurately analyze their trends, magnitudes, and drivers[17]. Whether GLOF activity is intensifying, stable, or diminishing requires further quantitative analysis[18,19]. Additionally, existing glacial lake risk assessment schemes in the Third Pole are primarily qualitative or semi-quantitative, and empirical or simple models are generally used to simulate GLOF paths and are combined with a relatively rough downstream exposure[20,21]. Although these first-order studies help to identify priority areas of concern on a large scale, they are insufficient to determine detailed GLOF risks and specific exposure locations for individual glacial lakes or local development planning[22].

Given these research gaps and the need to integrate and upgrade the current research essentials, we conduct a study to re-overview the glacial lake changes, GLOF characteristics, and risk assessment in the Third Pole. Ultimately, we map an inventory of glacial lakes and compile a GLOF dataset to reveal their changes and mutual relationships. Through numerical modeling and the construction of a detailed exposure dataset, we systematically assess and quantify the hazards, exposure, and risks of the glacial lakes in the Third Pole.

## Results

### Heterogeneous changes in glacial lakes

All of the glacial lakes with areas ≥0.02 km² and that are primarily fed by contemporary glacier meltwater were identified and included in the inventory using Sentinel-2A/B images acquired in 2018, 2020, and 2022. Classification was performed based on the topological positions of the lakes relative to their glaciers (see "Methods"). A total of 5894 glacial lakes were mapped in the Third Pole in 2022, with a combined area of 748.79 ± 41.16 km² and a combined volume of 20.13 ± 17.12 km³ (Fig. 1a). Among them, 869 lakes were classified as proglacial lakes (207.33 ± 8.2 km²; 10.37 ± 9.41 km³), 2735 as periglacial lakes (222.24 ± 15.91 km²; 4.84 ± 4.25 km³), 1929 as extraglacial lakes (290.22 ± 14.74 km²; 4.47 ± 3.123 km³), 113 as supraglacial lakes (6.92 ± 0.71 km²; 0.07 ± 0.05 km³), and 248 as ice-dammed lakes (22.08 ± 1.6 km²; 0.38 ± 0.27 km³). Periglacial or extraglacial lakes are the main types of glacial lakes in most of the regions, accounting for

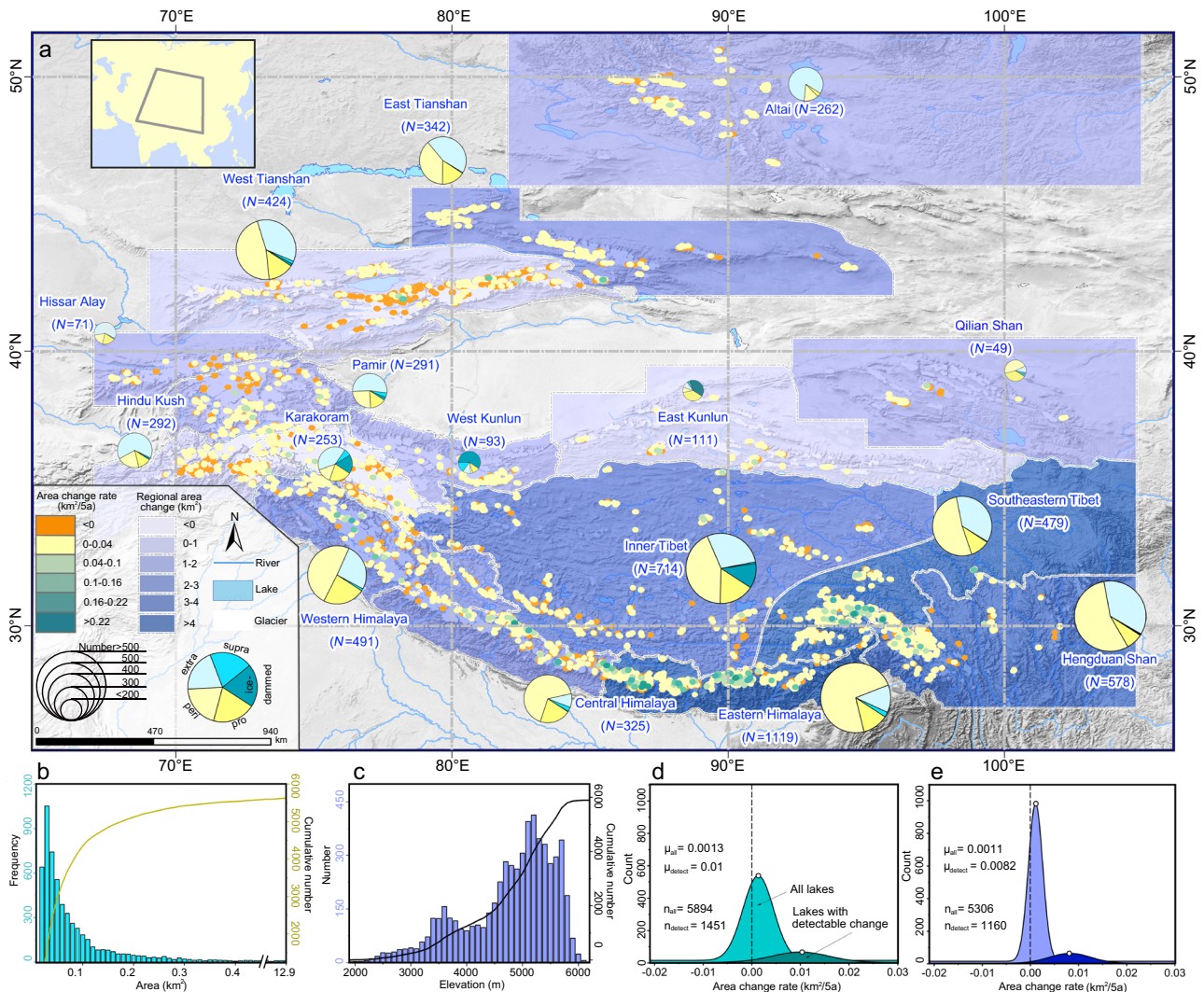

**Fig. 1 | Glacial lake distribution and changes in the Third Pole. a** Maps highlighting the glacial lake expansion rate between 2018 and 2022. The pie charts with different colors and sizes illustrate the number and types of glacial lakes based on the Global Terrestrial Network for Glaciers (GTN-G) regions. The histograms present the frequency of the glacial lakes in terms of their size (**b**) and elevation (**c**). The smoothed density distribution of the area change rate for all of the glacial lakes and lakes with detectable changes (error > change area) in 2018–2022 (**d**) and 1990–2018 (**e**).

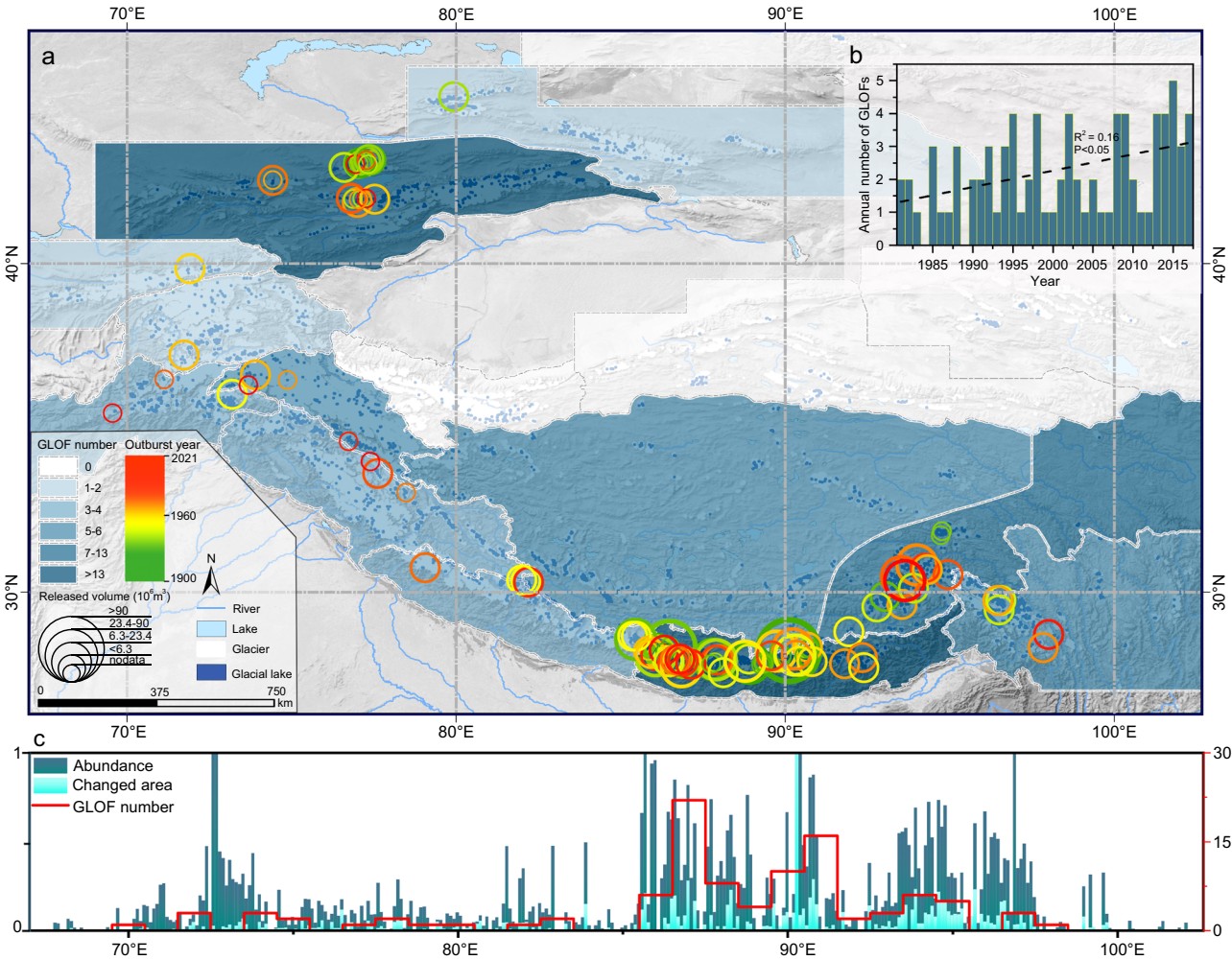

**Fig. 2 | Distribution of GLOFs in the Third Pole. a** Map depicting the spatial and temporal distributions of 145 GLOFs in the Third Pole since 1900. The sizes of the hollow circles represent the drainage volumes of the GLOFs, which were reconstructed based on observed or estimated values (Tables S2 and S3). The colors of the circles denote the outburst years of the GLOFs. **b** A relatively complete subset of GLOFs occurring during 1981–2020, which was used to analyze the GLOF frequency and trends. **c** The regional abundance and area changes of the glacial lakes, except for those in the Altai, Tianshan, and Qilian mountains, were normalized and summarized within 0.1° longitude intervals. The regional GLOF number was counted at 1°. The glacial lake number and expanded area factors were employed to depict the regional abundance and area changes of the glacial lakes, respectively. Their values were normalized to a range of 0–1.

an average proportion of 79%, except in East and West Kunlun where ice-dammed lakes predominate. Proglacial lakes directly connected to glaciers contribute an average of 17% of the total number of glacial lakes in all of the regions. Glacial lakes with a small size ($\leq$0.1 km$^2$) account for 74% of the total number of glacial lakes in the Third Pole (Fig. 1b). There are two altitude concentration intervals of 3400–3900 m and 4700–5800 m based on the number of glacial lakes (Fig. 1c), which are controlled by the increasing glaciation limits from southeast to northwest in the Third Pole.

Overall, the expansion of glacial lakes in the Third Pole has continued in recent years. Between 2018 and 2022, the area of the glacial lakes increased by $22.68 \pm 8.84$ km$^2$ (3.03% relative area change), and 83% of this total increase was contributed by proglacial lakes. The mean expansion rate of individual proglacial lakes ($0.022 \pm 0.002$ km$^2$/5 a) was an order of magnitude higher than that of the other lake types. To trace and compare the glacial lake changes since 1990, we combined the previously available inventories[7,16,23]. The results indicate that the total glacial lake area expansion rate in the Third Pole during 1990–2018 ($31.59 \pm 4.03$ km$^2$/5 a) was higher than the average level observed during 2018–2022 (Fig. S1). However, the expansion rate of the proglacial lakes increased from $13.82 \pm 1.33$ km$^2$/5 a in 1990–2018 to $18.89 \pm 2.12$ km$^2$/5 a in 2018–2022. Whereas 75% of the glacial lakes in

the Third Pole exhibited neutral changes (error > change area), recent observations have revealed a higher occurrence of both positive and negative changes compared to the period from 1990 to 2018 (Fig. 1d, e). As glacial lakes detach from their parent glaciers and reach their accommodation limit, their area and volume cease to increase and gradually diminish due to continuous sediment filling. Evidently, the growth of glacial lakes that become decoupled from their glaciers further slows[8]. The rapidly expanding glacial lakes were mostly proglacial lakes, and they were concentrated in the Eastern Himalayas, Southeastern Tibet, and Hengduan Shan, with significant regional differences compared to the rest of the Third Pole. In general, these proglacial lakes led the main expansion of the glacial lakes in the Third Pole in the last 5 years.

**Enhanced GLOF activity**

We compiled an inventory of GLOFs in the Third Pole by integrating various regional datasets (see Methods). We focused on GLOFs originating from moraine-dammed lakes (proglacial and periglacial lakes in this paper) since 1900. In total, we included 145 GLOFs from 122 lakes (Fig. 2a and Table S1). In particular, 93 of these events occurred after 1980 and were identified through the meticulous examination of documents and geomorphic features to better understand their

characteristics during the satellite era. The results show that there has been a significant increase in GLOF frequency since the 1980s (Fig. 2b), with the mean annual number increasing from 1.5 on average during 1981–1990 to 2.7 during 2011–2020. Spatially, GLOF activity has enhanced in Southeastern Tibet and China-Nepal border area, where 13 GLOFs occurred after 2010, surpassing the numbers in other regions of the Third Pole. The results of these trend analyses differ from those of previous studies that reported a decreased or unchanged GLOF frequency[18,24,25], demonstrating the necessity of dataset updates and integration. Globally, the reported number of GLOFs increased by at least 2.3 times from 2016 to 2023[26,27]. Further research on the triggers, trends, and climatic drivers of GLOFs remains an open and essential area of interest.

Ice avalanches and glacier calving are recognized as the main triggers of GLOFs worldwide[28,29]. Our inventory identified 68 triggers for GLOFs, excluding those in the Tianshan Mountains. Among them, 63% were triggered by ice avalanches, 9% by landslides, 10% by melting of buried ice in moraine dams, 15% by high temperatures and/or heavy precipitation, and 3% by upstream floods. The GLOFs in the Tianshan Mountains exhibited a distinctive outburst mechanism. They were generally released from short-lived glacial lakes and were triggered by temporarily opened ice tunnels[30–32]. These GLOFs had an average drainage volume of $0.28 \times 10^6$ m$^3$ (Fig. S2a and Table S3), which was significantly lower than that in the Eastern Himalayas ($6.42 \times 10^6$ m$^3$) and Southeastern Tibet ($4.37 \times 10^6$ m$^3$).

GLOF activity may indicate some degree of regional environmental instability. While the area change rate of an individual glacial lake has a low relevance with its GLOF susceptibility[33], it is undeniable that the regional GLOF activity is correlated with both the abundance of glacial lakes (Spearman's correlation coefficient $r = 0.87$) and their area change rate ($r = 0.86$) (Fig. 2c). This implies that these two factors may be suitable for evaluating the regional GLOF potential both now and in the future. In particular, the projected rapid glacier retreat and the distribution of numerous potential glacial lakes in the Third Pole suggest that the GLOF frequency would increase in the future[34,35].

## GLOF susceptibility and impacts

Understanding the current hazard posed by glacial lakes is of utmost importance. We developed a conceptual model and implemented a quantitative hazard assessment for all of the glacial lakes in the Third Pole, excluding the supraglacial and ice-dammed lakes (see "Methods"). We applied the conceptual model to evaluate 72 lakes that have previously experienced outbursts, and found 67 lakes were identified as posing a very high or high hazard before their outburst (Fig. S3). This 93% accuracy leads to a high level of confidence in determining potentially dangerous glacial lakes on a large scale. Among the 5535 glacial lakes assessed, a total of 379 glacial lakes were classified as having a very high hazard level and 1120 were classified as having a high hazard level (Fig. 3a).

For the lakes with a high outburst potential, we utilized the Hydrologic Engineering Centers River Analysis System (HEC-RAS) hydraulic model to simulate their potential GLOF impacts in a case-by-case manner. By combining the downstream maximum water depth and velocity of the GLOFs with the GLOF susceptibility indexes (Fig. 3b–d), the spatial hazards of the glacial lakes with a high outburst potential were mapped (Figs. 3e and S4). The results show that these GLOFs could inundate an area of ~6353 km$^2$. Inner Tibet has the largest potential inundation area of ~1513 km$^2$, followed by 850 km$^2$ in the Eastern Himalayas and 638 km$^2$ in the Western Himalayas (Table S5). This distribution is mainly controlled by the topography, as Inner Tibet has a flatter downstream debris fan or a relatively wide channel. On the national scale, China has the largest potential inundation area of 4080 km$^2$, which is an order of magnitude higher than those of other countries (Table S6).

We define an indicator of the GLOF probability to reveal the degree of GLOF threat at specific locations (see "Methods"). There are 28 valleys with a GLOF probability greater than 0.24, indicating the potential for more than five GLOF impacts from different sources. Most of them are concentrated in the Eastern Himalayas and Southeastern Tibet. Interestingly, eight out of the 28 valleys are located in transboundary basins, such as the Poiqu, Pengqu, and Gyigongzangbu river basins in the China–Nepal border area, which have maximum GLOF probabilities of 0.71, 0.62, and 0.71, respectively (Table S7). Furthermore, we identified 112 potential GLOFs with transboundary threats, including 55 along the China-Nepal border and 28 along the China-Bhutan border.

## Potential disaster volume

Based on GLOF simulation results, we extracted the detailed exposed elements along the GLOF paths and conducted a risk assessment (see "Methods"). The results provide scientific suggestions for local socioeconomic development planning and disaster prevention and mitigation strategies on both a large scale and a case-by-case basis.

Of the assessed 1499 glacial lakes with high outburst potentials, we identified 85 lakes as posing a very high risk and 113 as posing a high risk. There are 1228 glacial lakes that pose a downstream risk, raising concerns about the widespread potential of GLOF damage caused by glacial lakes in the Third Pole. The Eastern Himalayas have the highest risk level, which is twice as high as that in Southeastern Tibet, four times higher than that in the Central and Western Himalayas and Hengduan Shan, and more than eight times higher than that in West and East Tianshan, Inner Tibet, and Hindu Kush. The Poiqu River Basin has seven glacial lakes with a very high risk level, including the top-ranked Gangxico (28°21′N, 85°52′E) and the fourth-ranked Galongco (28°19′N, 85°50′E). These findings are consistent with those of previous studies conducted in the Third Pole[20,35] or its subregions[36,37].

In terms of the disaster-bearing bodies, ~55,808 buildings, 105 existing or planned hydropower projects, 194 km$^2$ of farmland, 5005 km of roads, and 4038 bridges are threatened by the potential GLOFs (Fig. 4a). By utilizing regional population distribution data, we estimated that roughly 190,000 lives are directly exposed within the GLOF paths. The regions with the highest exposure levels are the Eastern Himalayas, Southeastern Tibet, and West Tianshan, each of which have more than 10,000 buildings exposed in GLOF paths, which is 2–3 times the number exposed in the Hengduan Shan and the Central and Western Himalayas and is 4–7 times the number exposed in the East Tianshan, Hindu Kush, and Inner Tibet. The Himalayas have the largest number of exposed hydropower projects, posing concerns about secondary disasters from GLOFs[38,39]. Beihu (28°18′N, 86°09′E), Galongco, and Gangxico in the Poiqu River Basin have very high exposure, with impact lengths >110 km across China and Nepal. On the national scale, China has the highest exposure level, accounting for 46% of the exposed buildings in the Third Pole, 12% of the hydropower projects, 64% of the farmland, 59% of the roads, and 53% of the bridges. These values are 2.9 times higher than the numbers in Kazakhstan, 4.5 times higher than the numbers in Nepal, 6 times higher than the numbers in India and Pakistan, and 9.5 times higher than the numbers in Bhutan. It is noteworthy that the biggest city in Kazakhstan, Almaty (43°13′N, 76°54′E), is also threatened by GLOFs originating from the southern Ile Range, endangering over 6000 buildings.

## Regional GLOF characteristics

We calculated the distance between each glacial lake and the nearest human community downstream, and determined the basic early warning time for each dangerous glacial lake. Our findings reveal that the exposures in Southeastern Tibet and the Eastern Himalayas are in closer proximity to their respective glacial lakes, with median distances of 7 and 9.5 km, respectively, compared to other regions where

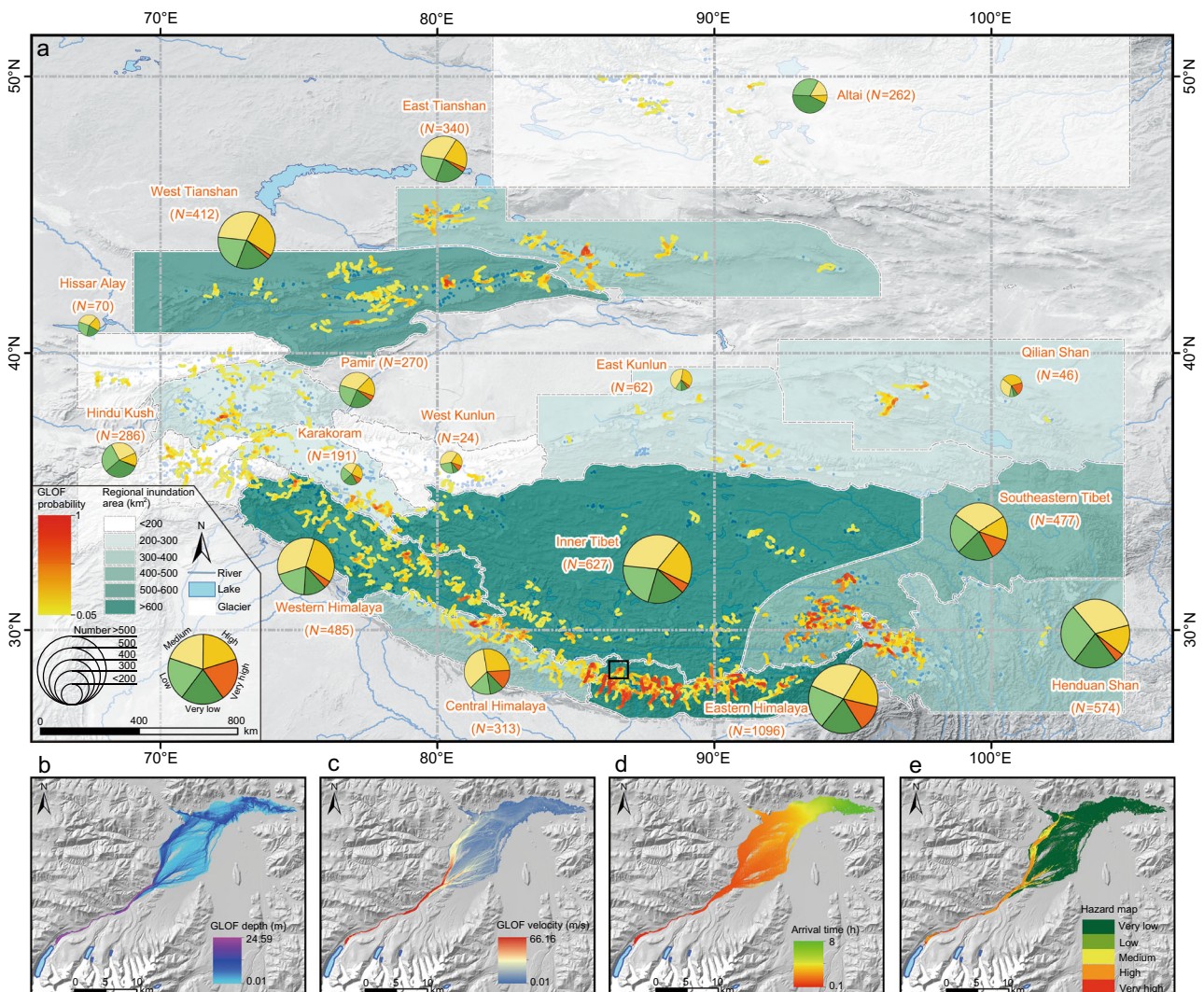

**Fig. 3 | GLOF susceptibility and simulation in the Third Pole. a** Composite map illustrating the results of the glacial lake hazard assessment, regional potential GLOF inundation area, and GLOF probability. The pie charts with different sizes and colors exhibit the numbers and classification of the glacial lakes used in the assessment processes. The potential flood propagation of each glacial lake with a high outburst potential was simulated using the HEC-RAS 2-dimensional hydraulic model. Colangma Lake was selected as an example to display HEC-RAS simulation results of maximum depth (**b**), maximum velocity (**c**), and arrival time (**d**), as well as a composition map of hazard (**e**). The black box in (**a**) shows the location of Colangma Lake.

the median distance ranges from 10 to 15 km (Fig. 4b). As a result, the median values of the basic early warning time in Southeastern Tibet and the Eastern Himalayas are 0.6 and 0.7 h, respectively, while those in other regions vary from 1 to 1.6 h (Fig. 4c).

We defined the potential disaster intensity as an indicator of the exposure level per unit area affected by GLOFs. The Hindu Kush has the highest regional average potential disaster intensity (0.516), followed by Hissar Alay (0.479) and Southeastern Tibet (0.383) (Fig. 4a). This implies that these regions may experience GLOFs characterized by small outbursts with high destructiveness in the future. These results emphasize the importance of the western Third Pole in the GLOF threat. Currently, the number and area of the contemporary glacial lakes in these regions are smaller, and the GLOF frequency and magnitude are much lower compared to those in the Eastern Himalayas and Southeastern Tibet. However, when the dramatic potential for the development of glacial lakes in the western Third Pole under future climate change scenarios is considered[34,35,40], the trend of the hazard and risk extension from east to west will be supported by the abundance of glacial lakes and downstream exposure.

## Discussion

This study demonstrates the changing status of glacial lakes in the Third Pole, as well as the characteristics of historical GLOFs and the potential threats posed by GLOFs. Through a combination of high-precision modeling and detailed downstream exposure data, an assessment was conducted to evaluate the GLOF hazards, exposure, and risks by employing an almost case-by-case methodology. Given their historical significance as major GLOF hotspots since 1980, the Eastern Himalayas and Southeastern Tibet remain the primary focus, and recent evidence suggests the intensification of GLOF activity due to the rapid expansion of glacial lakes in these regions. While the western region of the Third Pole, such as the Hindu Kush and Tianshan Mountains, currently exhibits a limited number of glacial lakes and slow expansion rates, the underestimated exposure elements raise concerns regarding the potential occurrence of small outbursts with high destructiveness in the future. The Tibetan Plateau and surrounding Tianshan, Hindu Kush, and Himalayas represent transboundary, densely populated, and water-demand strained mountainous areas, making them highly sensitive to political conflicts, climate changes, and natural hazards[1]. Our findings underscore the

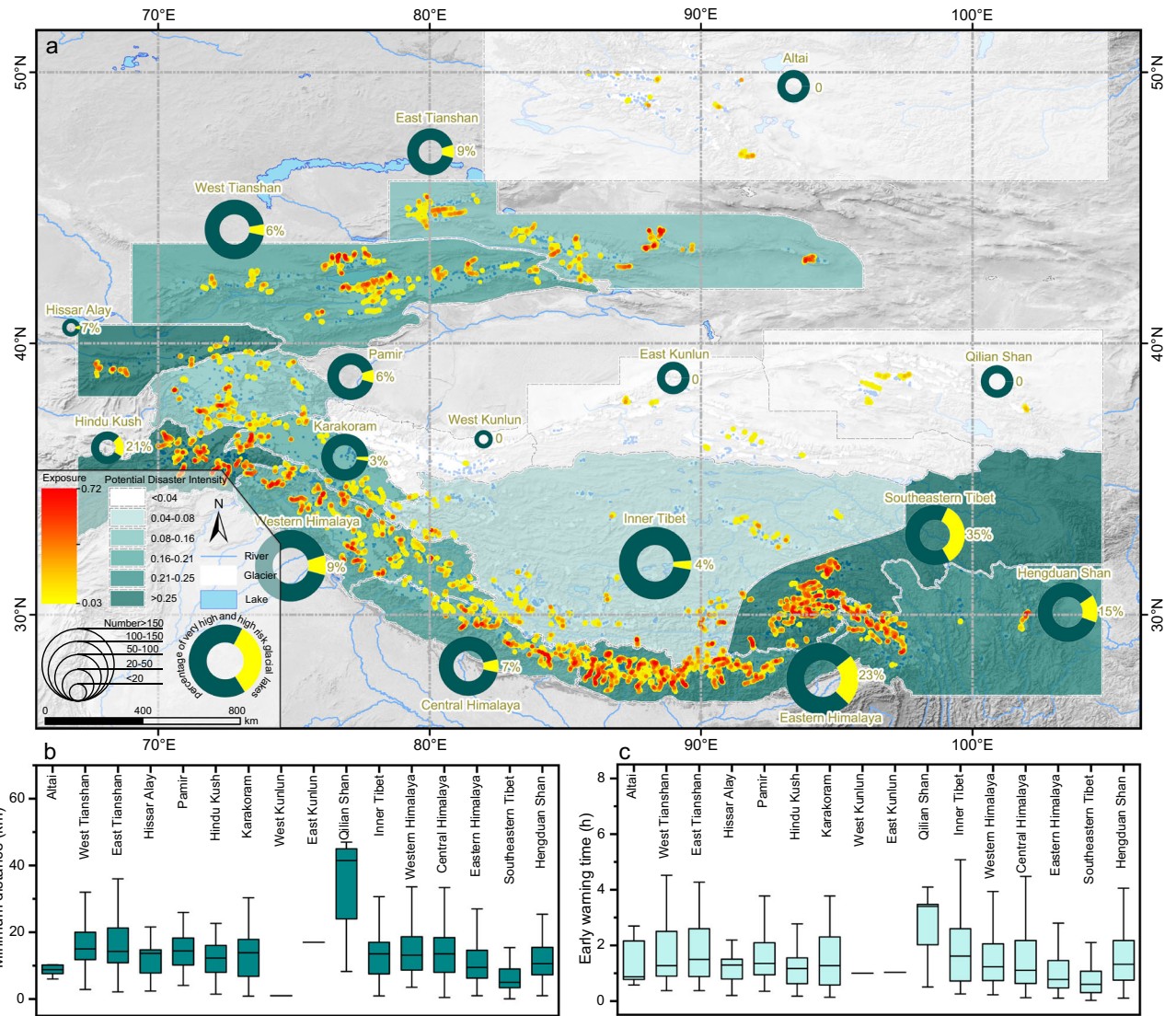

**Fig. 4 | GLOF exposure and risk in the Third Pole. a** Amplified map showing the spatial distributions of the GLOF risk, exposure, and regional potential disaster intensity. The yellow sections of the pie charts reflect the percentage of very high and high risk glacial lakes. **b** Box plots illustrating the distance between each glacial lake and the nearest human settlement downstream that is exposed, aggregated by region. **c** The basic early warning time of GLOFs aggregated by region.

significant challenges posed by the substantial potential disaster volumes in these economically disadvantaged and highly vulnerable regions. Considering the projected extension of GLOF threats toward the western regions under future climate change scenarios, it is crucial for the relevant nations surrounding the Third Pole to recognize the urgency of addressing GLOF threats and to promote regional cooperation.

Recently, two typical remedial works for preventing and mitigating GLOF threats have been implemented in the Poiqu River Basin in the Himalayas. First, in 2020, artificial drainage and dam reinforcement measures were implemented for the Jialongco Lake in response to the increasing GLOF hazard and rapid urban development in Nyalam County (28°09′N, 85°58′E). These measures resulted in a 50-m reduction in the water level[41]. Such measures can effectively reduce the potential magnitude and destructive impact of outburst floods when cities are directly exposed to GLOF threats[42]. Our assessment indicates that many cities/counties, such as Pulan (30°17′N, 81°10′E), Yadong (27°29′N, 88°54′E), and Langkazi (28°57′N, 90°23′E) in China, are at risk from GLOFs. In addition to pinpointing these GLOF threatened cities/counties, our study has exceptional implications for the planning and construction of

critical infrastructure projects, such as railways and hydropower plants. Evaluating the hazards and exposures of related glacial lakes, as well as assessing the likelihood and intensity of GLOFs at specific locations, is paramount for the effective implementation of engineering prevention and mitigation measures. Further evaluation in such areas would be meaningful.

Second, a monitoring and early warning system was established for the Cirenmaco Lake in 2020–2022. This glacial lake experienced a catastrophic transboundary GLOF in 1981, causing over 200 fatalities in China and Nepal[43,44]. Due to the increased hazard posed by the lake in recent years, research institutions have constructed a fully automated early warning system, enabling real-time data transmission, analysis, and alerts in alpine areas[1]. The installation of such systems should be prioritized in glacial lakes with high risk levels and situated away from downstream human settlements. The Himalayan region, in particular, faces a significant transboundary threat of GLOFs. The relatively large extent of the GLOF impacts and the limited time available for issuing warnings necessitate the development of monitoring, early warning, and data-sharing systems covering the upper, middle, and lower reaches of critical glacial lake basins. Yet, such systematic GLOF early warning projects are still lacking. Therefore, it is

necessary to enhance cross-regional cooperation to implement such critical early warning projects.

Currently, the implementation of remedial work in the Third Pole remains limited compared to that in the Peruvian Andes and the European Alps[45,46]. Our study provides scientific guidance for prioritizing hazardous glacial lakes and formulating further remedial works. The hazard maps generated for each dangerous glacial lake, along with the precise identification of downstream exposed communities and infrastructure, can be considered in local socio-economic development planning. Our investigation of GLOF risks in the Third Pole, conducted using a case-by-case methodology, yielded several high-quality datasets. This enables us to provide credible glacial lake assessment results and recommendations at different research scales, e.g., for the entire plateau, specific basins, and individual glacial lakes. These findings are of great significance for raising awareness and preparedness regarding the current GLOF threats in the Third Pole, as well as for future prognosis and sustainability. Furthermore, our evaluation methodology can serve as a valuable model for conducting repeated studies in other areas threatened by GLOFs worldwide, thereby improving our understanding of the global GLOF risk.

## Methods

### Glacial lake mapping and change analysis

Significant variations exist in the number of glacial lakes among the previously published inventories at the Third Pole scale[7,16,23,47,48]. These differences can be attributed to varying area thresholds of 0.05–0.003 km$^2$, as well as different definitions of glacial lakes. Initially, glacial lakes were identified as those within a 10 km buffer of a glacier and with a hydraulic connection to a glacier[23]. However, recent studies have included all lakes within the buffer regardless of glacier connections, resulting in inflated inventories. In this study, we needed to not only integrate previous research to analyze the state of the glacial lakes in the Third Pole but also to conduct a detailed risk assessment. We focused on glacial lakes with areas ≥0.02 km$^2$ that were primarily fed by contemporary glacier meltwater within a 10 km glacier buffer. Numerous thermokarst lakes and lakes without parent glaciers were excluded. Three time windows, 2018, 2020, and 2022, were selected to create new inventories and were combined with other available datasets to reveal the short-term and long-term glacial lake changes. A total of 878 Sentinel-2A/B images (10-m resolution) were used to manually delineate the glacial lakes (Fig. S5). To ensure a sufficient storage period for the glacial lakes during the year and to minimize the presence of mountain shadows, priority was given to images captured between July and November with less than 10% cloud coverage. The images were processed using a false-color composition of bands 4, 3, and 2 to highlight the water bodies. The initial locations for the lake extraction were based on the 2018 glacial lake inventory created by Wang et al.[7]. Throughout the workflow, each glacial lake underwent thorough review at least six times, ensuring the inventory's completeness according to our standards.

The glacial lakes were classified into five types based on their topologic positions relative to their glaciers (for example, see Rick et al.[49]): (1) proglacial lake, located in contact with the glacier terminus[50]; (2) periglacial lake, decoupled from its glacier but situated at the glacial moraine[51]; (3) extraglacial lake, located far from the glacier terminus and often with a landslide dam or without any dams[52]; (4) supraglacial lake, located on the surface of a glacier; and (5) ice-dammed lake, formed when a glacier surge blocks a downstream valley or when meltwater fills a depression between a retreating tributary and the main glacier (Fig. S6). This classification method emphasizes the connections between lakes and glaciers, facilitating the analysis of glacial lake dynamics within the context of glacial changes, compared to categorizing them based on the dam materials. Supraglacial lakes are typically small, exhibit significant fluctuations within and between years, and frequently appear and disappear[53,54]. In comparison,

numerous, occasionally concealed supraglacial lakes are another cause of the discrepancies in glacial lake datasets[47]. Our inventory only includes a small number of stable lakes. Additionally, our inventory included ice-dammed lakes that were overlooked by previous studies. These lakes are distributed across several ice caps in Inner Tibet and the Kunlun Mountains. Because they are often frozen, automatic identification approaches based on the normalized difference water index face challenges in extracting them The glacial lake volume was estimated using a set of empirical equations proposed by Zhang et al.[55], and its uncertainty was calculated by using 95% confidence intervals of the regression curves. The uncertainty in the glacial lake area ($\sigma$) was estimated as follows (Fig. S7):

$$\sigma = 0.5P \cdot I \tag{1}$$

where $P$ is the lake perimeter, and $I$ is the pixel resolution of the imagery. The absolute area change rate ($R$) was used to reflect the glacial lake expansion or shrinkage, and its uncertainty was estimated as follows:

$$R = (A_2 - A_1)/T \tag{2}$$

$$\sigma R = \sqrt{\left(\frac{\sigma A_1}{A_1}\right)^2 + \left(\frac{\sigma A_2}{A_2}\right)^2} \tag{3}$$

where $A_1$ and $A_2$ are the lake area in the first and second periods, respectively; $T$ is the time difference; and $\sigma A_1$ and $\sigma A_2$ are the area uncertainties. We calculated the glacial lake change error ($\sigma A$) as follows:

$$\sigma A = \sqrt{\sigma A_1{}^2 + \sigma A_2{}^2} \tag{4}$$

If error > |area change|, the lake change is not significant; otherwise, the lake exhibits a positive or negative change. To accurately calculate and interpolate the lake change rate over the past decades, especially for newly formed glacial lakes, we utilized the dataset for 1990 created by Wang et al.[7], that for 2000 created by Zhang et al.[23], and those for 2008, 2010, and 2015 created by Chen et al.[16]. However, due to variations in the study scales, glacial lakes in the Tianshan Mountains and Altai were often overlooked. To address this issue, we used 197 Landsat scenes (30 m resolution) to include these glacial lakes in the datasets. Overall, our relatively compact inventory can illustrate the underestimated expansion of the glacial lakes compared to previous inventories. For instance, Wang et al.[7] extracted data on all lakes within 10 km of glacier buffers and reported an average increase of 15.2% in glacial lake area during the period 1990–2018 in the Third Pole, whereas our inventory reveals an increase of 30.5%.

### GLOF characteristics

The GLOFs that occurred after 1900 and originated from moraine-dammed lakes were compiled (Table S2) by integrating several available GLOF datasets, such as those created by Zhang et al.[13], Shrestha et al.[14], and Veh et al.[25]. Many outburst floods produced by ice-dammed lakes or those without reliable data were not included (for example, Zheng et al.[24]). Benefiting from our extensive research on glacial lakes in the Third Pole, relatively accurate first-hand GLOF characteristics such as the triggers, drainage volume, and peak discharge were recorded to the fullest extent possible. The GLOFs that occurred after the 1980s were further validated and analyzed using satellite images[56]. First, we successfully reconstructed the drainage volumes for 65 drained glacial lakes by extracting the area differences before and after the GLOFs from 117 Landsat images and based on the relationship between the lake volume and area (Figs. S2a, S9a and Table S3). Second, based on the statistical data, we established a credible nonlinear

relationship between the peak discharge and drainage volume of the GLOFs (Fig. S2b and Table S4). These works contributed to the understanding of the regional GLOF characteristics and were utilized in subsequent simulations of potential GLOF propagations.

## GLOF susceptibility

Our evaluation sample consists of 5535 proglacial, periglacial, and extraglacial lakes, which were primarily dammed by a moraine or landslide. Supraglacial and ice-dammed lakes were not included in our evaluation scheme since they have different outburst mechanisms and are not among the main types in our inventory. Generally, their ice dams are broken due to dam flotation or ice tunnel enlargement[57]. This highlights the need for a separate evaluation system tailored to these lakes. Moreover, GLOFs originating from ice-dammed lakes exhibit significant regional clustering and periodic drainage patterns in the Third Pole[58,59]. By directing our attention to specific basins or ice-dammed lakes with well-defined issues, such as the Kaygar, Shishper, and Merzbacher lakes[60–62], we can effectively address their hazard and risk responses. Therefore, large-scale assessments of ice-dammed lakes appear to offer comparatively limited utility when compared with the more prevalent disasters and widely distributed hazard sources associated with the moraine-dammed lakes in the Third Pole.

The GLOF hazard signifies both the likelihood of a flood occurring (GLOF susceptibility) and its downstream impact. The methodology for assessing the GLOF susceptibility has been fully developed in recent years, achieving a transition from localized to large-scale, objective, and automated evaluation[20,63]. The key components of the assessment process include the selection of efficient indicators, weighting schemes, classification methods, reliability validation, and sensitivity analysis. Globally, numerous susceptibility evaluation studies have been conducted using qualitative, semi-quantitative, or quantitative methods, resulting in over 50 different evaluation indicators[64]. These indicators can be classified into five subsets based on the characteristics of the parent glacier, lake watershed, moraine dam, surrounding topography, and the glacial lake itself (Fig. S8), providing a holistic depiction of the GLOF susceptibility[64]. Typically, the criteria for indicator selection need to remain unbiased, and should consider properties such as exhaustiveness, non-redundancy, and consistency[65]. Quantitative evaluation methods generally fall into two categories. The first approach integrates all indicators into a single metric and subsequently classifies susceptibility levels to assign scores and ranks to each glacial lake. The second approach assigns a susceptibility rating to each indicator, such as utilizing multi-criteria decision analysis to derive a final lake level (for example, see Kougkoulos et al.[65]).

In this study, we adopted and expanded upon the quantitative evaluation framework developed by Zhang et al.[63]. They initially identified 17 suitable indicators by setting preliminary criteria, such as data availability and accuracy, and non-redundancy. Subsequently, using an optimality analysis tailored to the Himalayan GLOF characteristics, they determined the best combination of indicators, including (1) the mean slope of the parent glacier, (2) the potential for a mass movement to strike a lake (for example, see Allen et al.[20]), excluding the glacier-influenced potential, (3) the mean slope of the moraine dam, (4) the watershed area, and (5) the lake perimeter. These indicators characterize a range of GLOF triggers, including ice avalanches, landslides or rockfalls, dam settlement or piping, heavy precipitation or upstream incoming water, and hydrostatic pressure. Considering the significant role of ice avalanches in triggering GLOFs and the decreased susceptibility of some glacial lakes due to glacier retreat in recent years[66], we added an additional indicator, namely, the (6) horizontal distance between the glacier terminus and lake, to further highlight the impacts of glaciers. This indicator has been widely used and is highly effective in susceptibility assessment[67–69]. It is generally assumed that ice avalanches do not affect glacial lakes when the

distance exceeds 1 km[70,71]. Extraglacial lakes, located far from the parent glacier and developed in a consolidated moraine, were excluded from the consideration of the dam metrics. Since earthquake-triggered GLOFs are rare in the Third Pole, indicators for assessing the impacts of earthquake were not included in the selection process. Furthermore, a sensitivity experiment conducted in the Himalayas and Southeast Tibet demonstrated the ambiguous role of the influence of climate changes on GLOF susceptibility. As a result, indicators related to these factors were not incorporated into the assessment.

To calculate the slope and determine the watershed area, we utilized the 30 m Advanced Land Observing Satellite Global Digital Surface Model (AW3D30). Each indicator was normalized to a range of 0–1 and was assigned a weight using the analytical hierarchy process (AHP). While like AHP and some other multi-criteria decision analysis methodologies are prevalent in Earth Science, for relatively complex assessment cases, it should be applied cautiously, with input from experts in operations research to ensure the use of the most current and appropriate methodology. Based on the natural Jenks approach in ArcGIS, the final composited indexes of the GLOF susceptibility were classified into five categories: very low, low, medium, high, and very high. Seventy-two glacial lakes that had previously produced GLOFs were included in our assessment samples and were used to validate the evaluation accuracy of the GLOF susceptibility.

## GLOF simulation and hazard mapping

A GLOF simulation scheme was implemented specifically for glacial lakes with high or very high susceptibility levels (i.e., a total of 1499 glacial lakes with such a high outburst potential) without considering all of the glacial lakes. The HEC-RAS 2-dimensional hydraulic model was selected to simulate the potential flood propagation. This model has successfully been used to reconstruct and evaluate many GLOF events worldwide[72–74], demonstrating its practicality and efficiency. By utilizing the 2-dimensional flow surface, as well as a dam-breach hydrograph and high-resolution digital elevation model (DEM), an unsteady flow simulation was conducted for each glacial lake with a high outburst potential.

It should be noted that the complete drainage of a glacial lake is size-dependent. The statistical data revealed that since 1980, there have been 29 small outburst glacial lakes (<0.1 km²) with an average failure volume of 85% of the total lake volume, and as the lake size decreases, the frequency of complete drainage increases. Conversely, 36 medium/large glacial lakes exhibited an average failure volume of 58% of the total lake volume, with larger lakes experiencing relatively smaller drainage volume portions. To maximize the GLOF simulation within a reasonable range, we utilized the upper limit curve of the scatter value of the drainage volume and glacial lake volume ($V_d = 2.01V_t^{0.65}$) to determine the potential maximum drainage volume of each glacial lake (Fig. S9a), and we designated all of the small glacial lakes as completely drained. Due to the limited applicability range of this curve, we set the maximum drainage volume of the glacial lakes as $20 \times 10^6$ m³, resulting in restrictions for a total of 76 large glacial lakes. Subsequently, based on the estimated drainage volume and peak discharge, we adopted a linear assumption for the increase/decrease in the flood discharge to generate a dam-breach hydrograph[75]. Previous case studies have demonstrated that this hypothetical approach does not significantly affect the results[72,73,76]. Additionally, we established a relationship between the GLOF impact distance and drainage volume[25] to determine the length of the two-dimensional flow surface, ensuring that the flood remained contained (Fig. S9b).

Regarding the DEM used, we developed a composited scheme using two available DEMs for the Third Pole. The High Mountain Asia (HMA) DEM (8 m) was given priority as it was generated from very high-resolution commercial satellite imagery[77]. The HMA DEM provides detailed ground features but has a limited coverage[78]. Therefore, in areas where the HMA DEM is incomplete or missing, we utilized the

Advanced Land Observing Satellite (ALOS) Phased Array type L-band Synthetic Aperture Radar (PALSAR) DEM (12.5 m) either for mosaicking with the HMA DEM or for direct use. All of the DEMs were subjected to depression filling and surface smoothing processes to warrant unobstructed water flow. Given the need for extensive GLOF simulations in the topographically complex Third Pole region, the model requires a robust and efficient set of parameters that allow the simulated flow to adapt to narrow valleys such as those in the Himalayas, as well as relatively flat surfaces such as Inner Tibet. Through preliminary sensitivity analysis, we determined the crucial parameters of the model to be as follows: the minimum computational unit is 30 s, Manning's coefficient is 0.06, and the computation interval is 5 s.

By employing the HEC-RAS model, we obtained the distributions of the inundation extent, maximum water depth and flow velocity, and the arrival time for each potential GLOF. We defined an indicator, namely, the GLOF probability ($P_{GLOF}$), as the degree of threat posed by the GLOF at specific locations. This indicator was calculated as follows:

$$P_{GLOF} = \frac{\sum n_i}{N} \tag{5}$$

where $n_i$ represents the number of GLOF events in a specific site, and $N$ corresponds to the maximum number of GLOF occurrences in a specific site across the Third Pole.

Furthermore, we integrated the water depth, flow velocity, and hazard index to create a hazard map for each glacial lake. The mapping approach, based on the concept proposed by Zhang et al.[79], allows us to inform ground-level planning and response actions. The improved formula is utilized in the computation:

$$H_s = S * D * V \tag{6}$$

$$H_r = \sum_{i=1}^{n} S_i * D_i * V_i \tag{7}$$

where $H_s$ is the hazard for a single glacial lake; $S$ is the value of the GLOF susceptibility index; $D$ is the simulated flood depth; and $V$ is the flow velocity. For regional hazard mapping, various single GLOF hazards are overlain and calculated. Similarly, each hazard map is divided into five classes (very low, low, medium, high, and very high).

## Exposure and risk

In various GLOF events, the impacts commonly reported include damage to buildings, hydropower projects, farmland, roads, and bridges[27]. These indicators were employed to quantify the GLOF exposure. All of the infrastructure features, except for hydropower information, were obtained from OpenStreetMap (https://www.openstreetmap.org/). However, OpenStreetMap has limited coverage of features throughout the Third Pole[80]. We checked these structures along our simulated GLOF paths, resulting in a significant number of buildings, roads, and farmland not being mapped, particularly in villages situated in gullies, such as those in Inner Tibet, Hindu Kush, and Central Asia. With the exception of Nepal, which had better coverage, the completeness of the rest of the Third Pole was less than roughly 5% (Fig. S10). Hence, we manually supplemented the structure features for the entire Third Pole. Considering the uncertainty of the modeled floodplain boundaries, a buffer distance of 200 m around the potential inundation areas of the GLOFs was incorporated to the extract exposure elements from high-accuracy Maxar Premium Imagery. Finally, we identified a total of 520,610 buildings, 1996 km² of farmland, 98,630 km of roads, and 6035 bridges within these buffers. The reconstructed exposure elements can accurately reflect the construction status of the Third Pole between 2016 and 2020.

We normalized each indicator to a range of 0–1 and employed the AHP method to allocate weights to the exposed buildings, hydropower projects, farmland, roads, and bridges. This ranking takes into account their varying levels of impacts on humans. By integrating these factors, we derived the exposure index for glacial lakes. The GLOF risk is the result of multiplying their normalized hazard by the exposure. Both the exposure and risk were also classified into five categories, i.e., very low, low, medium, high, and very high, utilizing the natural Jenks approach. To investigate the characteristics of the GLOFs at both the single lake and regional scales, we defined the level of exposure per unit of inundated area as the potential disaster intensity (PDI), which was calculated as follows:

$$PDI = \frac{E}{A_i} \tag{8}$$

where $E$ is the GLOF exposure, and $A_i$ is the normalized inundation area of the GLOF.

## Robustness of assessments and simulation

The indicators used to assess the glacial lake susceptibility have been carefully selected to correspond to various GLOF triggers, and weights have been assigned based on their importance. This contributes to a more scientifically grounded assessment. In the GLOF simulation scheme, glacial lakes without a high outburst potential were excluded. Previous studies have verified that over 90% of drained lakes can be identified as having a high or very high susceptibility[20,35]. Moreover, in comparisons of different assessment cases, glacial lakes with a high outburst potential can be robustly identified even using different combinations of evaluation indicators and weighting schemes[63]. These phenomena indicate that scaling down the sample for further GLOF simulations would not significantly impact the results as it covers all of the interesting glacial lakes and ensures the accurate determination of future GLOF sources. For example, our simulation samples include all eight important glacial lakes in Nepal identified by Rounce et al.[81] and ICIMOD[82], and our assessment of the GLOF threat to Almaty City is in agreement with the results of Bloch et al.[83].

Due to the nature of conducting a maximum potential GLOF simulation, quantifying uncertainties is challenging. There are two key sources of uncertainty to consider. The first is the assignment of model parameters. For example, we selected a relatively smaller Manning's value to enhance the water flow efficiency. A sensitivity analysis performed in the reconstruction of the 1981 Cirenmaco GLOF in the Poiqu River Basin indicated that variations in the Manning's value had no decisive impact on the flood inundation area and mean water depth[72]. This conclusion aligns with our modeling requirements. Second, the simulation accuracy of GLOFs is primarily determined by the high-resolution DEM as it serves as the main input data. There are essential differences between the results obtained from the commercial imagery produced HMA DEM and freely available PALSAR DEM. GLOF propagation is significantly hindered by extensive depression filling on the surface of the PALSAR DEM, leading to underestimations of the flood inundation area, arrival time, and downstream exposure. Of the 1499 GLOF propagation simulations, the HMA DEM was utilized for 52.6%, which guarantees the fundamental credibility of our results. The mosaic production was used in 30.7% of the simulations, and the ALOS PALSAR DEM was only used in 16.5% of the simulations. Regionally, the HMA DEM was less frequently applied in the Altai and Southeastern Tibet regions, limiting the reliability of their simulation results. Overall, it should be noted that our maximum potential GLOF simulation and exposure analysis may still underestimate the actual risk due to data limitations in various areas.

## Data availability

The Landsat images can be downloaded from the United States Geological Survey (USGS) website (https://earthexplorer.usgs.gov/). The Sentinel images can be downloaded at https://scihub.copernicus.eu/dhus/#/home. The Advanced Land Observing Satellite (ALOS) Global Digital Surface Model (AW3D30 v2.2) can be downloaded at https://www.eorc.jaxa.jp/ALOS/en/aw3d30/index.htm. The High Mountain Asia (HMA) digital elevation model (DEM) and ALOS Phased Array type L-band Synthetic Aperture Radar (PALSAR) DEM can be downloaded at https://search.earthdata.nasa.gov/search. The Randolph Glacier Inventory 6.0 data can be download at http://www.glims.org/RGI/. The data for existing and planned hydropower projects can be downloaded at http://globaldamwatch.org/grand/. The regional population distribution dataset in 2023 can be downloaded at https://ghsl.jrc.ec.europa.eu/ghs_pop2019.php. The inventories of glacial lakes in the Third Pole in 2018, 2020, 2022 generated in this study have been deposited in the Zenodo database under accession code https://doi.org/10.5281/zenodo.8369313. The simulated maximum water depth and velocity, arrival time, and hazard map produced for each potential GLOF in this study have been deposited in the Zenodo database under accession code https://doi.org/10.5281/zenodo.8369351. The exposure dataset generated in this study have been deposited in the Zenodo database under accession code https://doi.org/10.5281/zenodo.8369266.

## Code availability

We use HEC-RAS model to simulate glacial lake outburst flood. The HEC-RAS model can be downloaded at https://www.hec.usace.army.mil/software/hec-ras/.

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

## Acknowledgements

This study was supported by the Second Tibetan Plateau Scientific Expedition and Research (STEP) Program (2019QZKK0208), the Strategic Priority Research Program of Chinese Academy of Sciences (XDA20060201) and the International Partnership Program of Chinese Academy of Sciences (131C11KYSB20200029).

## Author contributions

T.Z. and W.W. designed the study, analyzed data, and wrote the draft of the manuscript. B.A. edited the manuscript. L.W. contributed to construct the exposure dataset.

## Competing interests

The authors declare no competing interests.
