## [Peer Review File · Nature Communications]

Enhanced glacial lake activity threatens numerous communities and infrastructure in the Third PoleEditorial Note: Parts of this Peer Review File have been redacted as indicated to remove third-party material where no permission to publish could be obtained.

REVIEWER COMMENTS

Reviewer #1 (Remarks to the Author):

The author's efforts in the study of glacial lakes and their outburst flood in the Third Pole are very commendable. As mentioned in the manuscript it is a comprehensive study which was lacking in the past in such a holistic approach. However, I found that the manuscript missed some basic things about the glacial lake study.

My first concern is the lack of detailed information about the HEC-RAS hydraulic modeling (Lines 141-142). It is good that the results give the area of inundation, distance traveled downstream and possible number of physical infrastructure damages. However, it is not mentioned how deep the breach occurs and how much water will come out from the lake that takes as input to the HEC-RAS. I strongly suggest giving detailed information on such crucial parameters of HEC-RAS.

My second concern is the lack of a literature review of related published papers and reports on the glacial lakes of Nepal. None of the papers and reports of the glacial lakes of Nepal listed the Tilicho Lake (lines 181-182) as a dangerous glacial lake but it was mentioned as the most dangerous glacial lake in Nepal. If so, I strongly request to mention the criteria for defining a lake as a dangerous glacial lake. Yes, it is the largest lake (4.8 sq. km) in the world lying at a high altitude (4,919 m). All large lakes at a high altitude do not pose always a risk. Therefore, the criteria for defining a lake as a dangerous glacial lake is essential. Otherwise, past criteria should be adopted with a proper reference. Again it mentioned that the impact length is 120 km downstream. For it, need to mention what breaching depth and volume of water could release from the lake. It is related to the first concern above.

My third concern is about Figure 4a (lines 176-178). It is not clear information about the hydropower and other infrastructures in the figure. Need to re-check.

Reviewer #2 (Remarks to the Author):

This research offers an exhaustive review of glacial lake transformations, GLOF attributes, and risk evaluation in the Third Pole. In essence, the authors created a detailed catalogue of glacial lakes and assembled an enhanced GLOF dataset to understand their evolutions and interconnections. By employing numerical simulations and building a comprehensive exposure dataset, they methodically evaluated the dangers, vulnerability, and potential threats posed by glacial lakes in the Third Pole. Despite the great efforts from the authors, who compiled large amounts of valuable data for the region and run numerous simulations in each phase of their analysis, I detected several major issues, primarily associated with their risk analysis datasets and methodology, that need to be addressed before this manuscript could be accepted for publication.

Major issues:

A) Is this a truly comprehensive assessment?

In the introduction (L41-45), the authors highlight a notable absence of a thorough dataset for glacial lakes in the specified region. They further suggest that existing knowledge might be influenced by biases and discrepancies, which I concur with. They also reference a study by Rick et al. (2022) (L358-362) that categorizes lakes into five distinct topological types. Given this context, one would anticipate a comprehensive susceptibility assessment encompassing all five lake types. This would ideally guide decision-makers in allocating resources based on the highest risk lakes. However, a subsequent section (L399-402) reveals that the authors chose to exclude two of these five types from their assessment. While I'm not necessarily advocating for the inclusion of these two types in their study, it's essential for the authors to provide clarity regarding which lake types their risk analysis refers to throughout their manuscript. Does it cover all the lakes in the region, or only a selective

subset?

B) Decision analysis theory and rules for criteria

The integration of operations research methods like AHP, FCM, and TOPSIS in geosciences is expanding, offering valuable insights into risk assessments and intricate decision-making. However, with these techniques originating from distinct scientific disciplines, geoscientists must apply them judiciously to avoid misconceptions and unsound practices. A key point to understand is that the AHP is primarily suited for ranking and choice problems, not for sorting (Ishizaka et al., 2012). This limitation implies that while AHP can rank the relative risks among alternatives (e.g., glacial lakes), it won't objectively determine the inherent risk of an individual lake. This distinction is crucial; otherwise, resources might be misallocated based on perceived risks (as evidenced in Kougkoulos et al., 2018).

When assessing GLOF susceptibility, the authors employ the Zhang et al. (2021) method that utilizes an FCM for weighting hazard criteria. While an intriguing approach, the rationale behind selecting FCM over other multi-criteria methods isn't clear. It would be beneficial to consult comprehensive works like Watrobski et al. (2018), which guide the selection of MCDA methods and could potentially enhance the robustness of their analysis. The same recommendation applies to the AHP technique employed later in the paper.

Additionally, there's a concern with the selection of criterion (2) at L416-417. For an MCDA to remain unbiased, three primary properties must be maintained: a) exhaustivity, b) cohesion, and c) non-redundancy. The current criterion (2) appears to fall short due to concealed sub-criteria. Briefly, the risk of mass movement impacting a lake is influenced by a) the surrounding slopes (as pointed out by Zhang et al. regarding non-glacierized areas with a slope $> 30^\circ$) but also by b) the proximity of these slopes to the lake. Consider two lakes: one surrounded by extensive steep terrain 500m away and another with a smaller steep area just 20m away. The modellers choice to score based solely on a) or to combine a) and b) together in a single criterion can skew results. To address this, the authors could introduce an additional criterion estimating the distance to non-glaciated slopes, especially since they've already incorporated a criterion for glacier-to-lake distance. I'd advise the authors to delve deeper into MCDA criteria selection theories, referencing key works like Hites et al. (2005), to enhance the robustness of their analysis.

Lastly, in L495-496 the authors use AHP and assign weights to the exposed infrastructure based on their importance. Could the authors please clarify how did they rank the importance? Is it based on expert judgment, personal judgment, or some actual data? At the moment it is not clear and this could change the final result, that is why I would appreciate if clarifications were given.

C) GLOF simulation and hazard mapping clarifications

This section is impressive. The fact that the authors simulated 1500 GLOF propagation simulations in HEC-RAS shows true determination for a comprehensive assessment. Therefore, I salute this effort. Nevertheless, some points are not very clear to me. I know that this remains a large-scale assessment and detail is sometimes difficult (running multiple scenarios for each outburst would prove to be a very laborious task), but there is no clear indication of the values used in the simulations. In L451-454 the authors indicate the shape of the hydrograph, but how was the manning's value calculated? How uncertain are the flood results since no sensitivity analysis was conducted (based on multiple flood scenarios)? What about flood duration? Also, I would appreciate more technical details on the simulation time, the hardware used to run the simulations etc. All this would fulfil the criterion of reproducibility which is very important in our science and would pave the road for future studies.

Minor comments:

L10: Change "cryosphere" to "cryospheric".

L21-23: Please rephrase since this sentence is quite long and confusing.

L26: Highly vulnerable to what?

L33-35: These are not the only two reasons causing an outburst flood.

L39: Is there any article in English illustrating these GLOFs?

L41-42: Consider citing here the large review study by Emmer et al. (2022) which grouped representatives from 17 different countries and identified and elaborated trends and challenges and proposed possible ways forward to navigate future GLOF research.

L111-114: These numbers add up to 97%. What about the remaining 3%?

L121: What exactly is GLOF intensity? Is it GLOF magnitude? Impacts from GLOFs? Flow velocity? Could you please explain.

L121-L122: Could this be redundant? What if it is just the glacial lake area in a region? The number of lakes would basically just increase the area covered in a region.

L137: ...and that found 67 were. Delete "that".

L176-190: I think it would be nice to see also an estimation of the number of people affected. Even if it is a rough estimate.

L240-241: There is a terminology confusion between hazard and risk here. I think that early warning systems should be a priority in glacial lakes that pose the highest risk. If a lake is only very susceptible to burst but doesn't have vulnerable elements downstream this ranks lower as a priority.

L357-363: The authors chose the classification by Rick et al. (2022) instead of classifications based on dam materials. I like the idea, but I would like to also see an argument behind this initiative.

L395: In Supplementary Table 3 (Reconstruction of drainage volume for the historical GLOFs after 1980), we can see the area before outburst (in km²) but not after the drainage. I think both should appear.

L490-491: I appreciate the effort of the authors to manually digitize features for the entire third pole. First, I would like to ask if it would be possible to add this data in supplementary material (e.g. in .shp or other format) which would largely benefit future studies in the region. Second, they claim to have used high accuracy Maxar Premium Imagery for this task. Nevertheless, in my knowledge this imagery is not open-access, and I don't see any grant ID in the data availability section. Could the authors provide more information on how they acquired this data. Also, it would be highly appreciated if they indicated why they used a 200m buffer.

Figures

Fig.1: Delete Westerlies and Indian Monsoon (it adds unnecessary detail to the map). For area change rate I would suggest a Green-Yellow-Orange-Red colouring.

Fig. 2: This map needs to be presented in another way. The circles and dates are impossible to distinguish.

Summary

This manuscript presents a commendable glacial lake risk assessment for the Third Pole. The authors' cross-disciplinary approach, the extensive data amassed, and the significant risk scoring findings clearly reflect their meticulous efforts. However, there are areas for important refinement, particularly regarding the clarity on the specific types of glacial lakes analysed, the choice of operational research methods like FCM and AHP, and the somewhat sparse detailing and protocol for the HEC-RAS simulations.

Please, do not hesitate to contact me in case of questions (ikougkoulos@acg.edu)

Kind regards

Ioannis Kougkoulos

References to be considered:

Emmer, A., Allen, S. K., Carey, M., Frey, H., Huggel, C., Korup, O., ... & Yde, J. C. (2022). Progress and challenges in glacial lake outburst flood research (2017–2021): a research community perspective. *Natural Hazards and Earth System Sciences*, 22(9), 3041-3061. <https://doi.org/10.5194/nhess-22-3041-2022>

Hites, R., De Smet, Y., Risse, N., Salazar-Neumann, M., & Vincke, P. (2006). About the applicability of MCDA to some robustness problems. *European Journal of Operational Research*, 174(1), 322-332. <https://doi.org/10.1016/j.ejor.2005.01.031>

Ishizaka, A., Pearman, C., & Nemery, P. (2012). AHPSort: an AHP-based method for sorting problems. *International Journal of Production Research*, 50(17), 4767-4784. <https://doi.org/10.1080/00207543.2012.657966>

Wątróbski, J., Jankowski, J., Ziemia, P., Karczmarczyk, A., & Ziolo, M. (2019). Generalised framework for multi-criteria method selection. *Omega*, 86, 107-124.

<https://doi.org/10.1016/j.omega.2018.07.004>

Reviewer #3 (Remarks to the Author):

General comments:

In recent years, glacial lakes have become a research hot topic in the field of Earth Science due to their rapid expansion on a global scale and the resulting many significant GLOF disaster events. Understanding the current glacial lake risks is important for predicting the future risks, and it is of great importance to evaluate the increasing GLOFs hazard threats worldwide.

This study systematically sorted out the characteristics of current glacial lake activity (including changes and GLOF characteristics) in the Third Pole region, and quantitatively assessed the risk of downstream exposure elements caused by potential GLOFs. Based on a combination of high-precision modeling and detailed downstream exposure data, a comprehensive assessment was conducted to evaluate the GLOF hazards, exposure, and risks by employing an almost case-by-case methodology in this manuscript. It is a widely focused scientific topic with potentially wide audience, and I am particularly persuaded that the scientific findings supported by largely amount data, especially for integrated hazard evaluation method and the unprecedented exposure data of GLOFs hazards. In general, the assessment method and concept of this study are novel, and the conclusions drawn are also reliable. So, overall good achievements, such as comprehensive investigation of GLOF risks results, methodology for potential application for global glacial lake hazards evaluation outside the Third Pole region, several yielded valuable datasets, make me recommend this manuscript for a potential publication in Nature Communications after necessary revisions.

However, the structure of the first part of the method "Glacial lake mapping and change analysis" needs some adjustments. After describing the reasons for the current differences in the list of each dataset, it is necessary to explain the extraction standard of the glacial lake in this study, and then elaborate how to extract the glacial lake from the image. A comparison needs to be added to explain the advantages of the definition of a glacial lake (excellent by a sketch map to illustrate the difference of definitions and connections to glaciers). Furthermore, I think the uncertainty appraisals and discussions of the calculated lake volume and simulated GLOFs (depth, velocity, arrival time, etc.) are incomplete or even missing. I strongly suggest that discussions of uncertainties should be strengthened.

Additionally, the authors set the minimum glacial lake threshold at 0.02 km² and excluded lakes within the glacier's 10 km buffer zone that were not hydraulically connected to the glacier, which made the dataset compact and suitable for evaluation. The authors should make comparison with previous studies to illustrate the differences and advantages between the glacial lake dataset in this study and the previous datasets. Multi sources and types of data with different resolutions are used from network, and some of them were examined or edited by the authors. The authors are also required to comprehensively describe their practicability, reliability and accuracy, and their possible or potential impacts on the results of GLOFs susceptibility, simulation, hazards mapping, exposure and risk.

Specific comments:

L65-69, Linguistic streamlining without too many repetitive sentences. I suggested providing a sketch map (extended figure) as extended Figure to illustrate the definitions and differences of five types of glacial lakes, including their connections to glaciers.

L100-101, more information about the relationships of moraine-dammed lake, proglacial lake and periglacial lake is needed. Here you only focused on moraine-dammed lakes or the proglacial and periglacial lakes that were moraine-dammed lakes?

L104-105, Is there likelihood that more GLOFs events are recorded because of the increasing capacity of satellite monitoring or other reasons? As it was also reported that no distinct GLOFs frequency was detected in Hindu Kush–Karakoram–Himalaya–Nyainqentanglha region (Veh et al., 2019).

L 115-116, What types these short-lived glacial lakes are? They are moraine-dammed lake? Why there exist ice-tunnels? I did not carefully read the documents of 28-30, however, in my knowledge, the trigger of ice tunnels usually appeared in ice-dammed lakes.

L121, The regional abundance of glacial lakes can be expressed using specific factors, such lake number, area, and volume, etc., please point out which factor used in figure 2c. The area change rate is in the same situation.

L 130-132, in Figure 2c, the unit of y-axis is indistinct. Are they in frequency or percentage?

L168, Please note that these 1499 glacial lakes belong to high or very high susceptibility.

L182, Add latitude and longitude to the first occurrence of a place or glacial lake name.

L 193-194, what the bars indicate in Fig.4b and c?

L204-211, Logical order may be problematic.

L 344, How do you determine and practically operate connections of glaciers and lakes?

L358, Note the complete use of brackets.

L 366-367, How about the uncertainty of the calculated lake volume?

L 417, Maybe, the indicator of mean downstream slope of the moraine dam is more pertinent than the mean slope of the moraine dam.

L 419-420, the indicator of horizontal distance between the glacier terminus and lake is not newly used. As far as my knowledge, at least it is used in Wang et al., 2008, 2012, 2013 where 0.5km and 1km were used as thresholds.

L486, Please add the weblink of OpenStreetMap.

L535, Some papers may be finished their discussion in AGU and accepted now, please write the complete reference information.

L603, Uniformity of color and size of symbols in this figure.

Reviewer #1 (Remarks to the Author):

The author's efforts in the study of glacial lakes and their outburst flood in the Third Pole are very commendable. As mentioned in the manuscript it is a comprehensive study which was lacking in the past in such a holistic approach. However, I found that the manuscript missed some basic things about the glacial lake study.

Reply: Thank you for acknowledging the significance of our study. By considering both the general and specific comments provided by the three reviewers, we have undertaken substantial revisions of this manuscript. This includes a detailed description of the HEC-RAS method alongside, an explanation of the selection standards of the evaluation indicators, and a discussion of the uncertainties of the assessments and simulations.

We provide point-to-point responses to the reviewers' comments below. The reviewers' comments and questions are in black font, and our responses are in blue font.

My first concern is the lack of detailed information about the HEC-RAS hydraulic modeling (Lines 141-142). It is good that the results give the area of inundation, distance traveled downstream and possible number of physical infrastructure damages. However, it is not mentioned how deep the breach occurs and how much water will come out from the lake that takes as input to the HEC-RAS. I strongly suggest giving detailed information on such crucial parameters of HEC-RAS.

Reply: We thank the reviewer for the positive comments and the helpful recommendations. The HEC-RAS two-dimensional hydraulic model was selected to simulate the potential flood propagation. This model has successfully been used to reconstruct and evaluate many GLOF events worldwide, demonstrating its practicality and efficiency (Anaconda et al., 2015; Wang et al., 2018). By utilizing the two-dimensional flow surface, as well as a dam-breach hydrograph and a high-resolution digital elevation model (DEM), an unsteady flow simulation was conducted for each glacial lake with a high outburst potential.

The drainage water of the glacial lake was estimated using an empirical equation which is determined by previous GLOFs (Supplementary Figs. 2, 9). It should be noted that the complete drainage of a glacial lake is size-dependent. To maximize the GLOF simulation within a reasonable range, we utilized the upper limit curve of the scattered data points of the drainage volume and glacial lake volume ($V_d = 2.01V_t^{0.65}$) to determine the potential maximum drainage volume of each glacial lake (Supplementary Fig. 9). Subsequently, based on the estimated drainage volume and peak discharge, we adopted the assumption of a linear increase/decrease in the flood discharge to generate a dam-breach hydrograph (Westoby et al., 2014). Previous case studies have demonstrated that this hypothetical approach does not significantly affect the results.

The simulation accuracy of the GLOFs is primarily determined by the high-resolution DEM as it serves as the main input data. Regarding the DEM used, we developed a composited scheme using two available DEMs for the Third Pole. The High Mountain Asia (HMA) DEM (8 m) was given priority as it was generated from very high-resolution commercial satellite imagery. The HMA DEM provides detailed ground features but has a limited coverage. Therefore, in areas where the HMA DEM is incomplete or missing, we utilized the Advanced Land Observing Satellite (ALOS) Phased Array type L-band Synthetic Aperture Radar (PALSAR) DEM (12.5 m) either for mosaicking with the HMA DEM or for direct use. All the DEMs were subjected to depression filling and surface smoothing processes to warrant unobstructed water flow. Of the 1,499 GLOF propagation simulations, the HMA

DEM was utilized in 52.6%, which guarantees the fundamental credibility of our results. The mosaic production was used in 30.7% of the simulations, and the ALOS PALSAR DEM was only used in 16.5% of the simulations.

Given the need for extensive GLOF simulations in the topographically complex Third Pole region, the model requires a robust and efficient set of parameters that allow the simulated flow to adapt to narrow valleys such as those in the Himalayas, as well as relatively flat surfaces such as Inner Tibet. Through preliminary sensitivity analysis, we determined the crucial parameters of the model to be as follows: the minimum computational unit is 30 m, Manning's coefficient is 0.06, and the computation interval is 5 s.

Plot of drainage volume versus the glacial lake volume

Anaconda, P. I., Mackintosh, A. Norton, K. Reconstruction of a glacial lake outburst flood (GLOF) in the Engano Valley, Chilean Patagonia: Lessons for GLOF risk management. *Sci. Total Environ.* **527–528**, 1–11 (2015).

Wang, W. et al. Integrated hazard assessment of Cirenmaco glacial lake in Zhangzangbo valley, Central Himalayas. *Geomorphology* **306**, 292–305 (2018).

Westoby, M. J. et al. Modelling outburst floods from moraine-dammed glacial lakes. *Earth Sci. Rev.* **134**, 137–159 (2014).

My second concern is the lack of a literature review of related published papers and reports on the glacial lakes of Nepal. None of the papers and reports of the glacial lakes of Nepal listed the Tilicho Lake (lines 181-182) as a dangerous glacial lake but it was mentioned as the most dangerous glacial lake in Nepal. If so, I strongly request to mention the criteria for defining a lake as a dangerous glacial lake. Yes, it is the largest lake (4.8 sq. km) in the world lying at a high altitude (4,919 m). All large lakes at a high altitude do not pose always a risk. Therefore, the criteria for defining a lake as a dangerous glacial lake is essential. Otherwise, past criteria should be adopted with a proper reference. Again it mentioned that the impact length is 120 km downstream. For it, need to mention what breaching depth and volume of water could release from the lake. It is related to the first concern above.

Reply: Thank you very much for the comment. We have reviewed the extensive assessment work carried out by previous researchers in the Nepal region and have cited related references (e.g., ICIMOD, 2011; Khadka et al., 2021; Rounce et al., 2016, 2017).

We highlight Tilicho Lake in previous version of manuscript because the lake has the high downstream exposure, rather than the lake is the most dangerous glacial lake in Nepal. We strongly agree with review's comment that "All large lakes at a high altitude do not pose always a risk". Our criteria for defining a lake as a dangerous glacial lake is described in Methods section (see GLOF susceptibility subsection). It includes selection of six indicators, (1) the mean slope of the parent glacier, (2) the potential for a mass movement to strike a lake, excluding the glacier-influenced potential, (3) the mean slope of the moraine dam, (4) the watershed area, (5) the lake perimeter, and (6) horizontal distance between the glacier terminus and lake. These indicators characterize a range of GLOF triggers, including ice avalanches, landslides or rockfalls, dam settlement or piping, heavy precipitation or upstream incoming water, and hydrostatic pressure. All the evaluation indicators used in this study were through a rigorous screening process. They are often used to assess GLOF hazard both in the Himalayas and elsewhere in the world.

Our assessment aligns well with prior research findings. For instance, our analysis of the glacial lakes with a high outburst potential includes all eight significant glacial lakes in Nepal that were identified by Rounce et al. (2016). These lakes are Chamlang North Tsho, Chamlang South Tsho, Dig Tsho, Imja Tsho, Lower Barun Tsho, Lumding Tsho, Thulagi Tsho, and Tsho Rolpa. In terms of GLOF susceptibility, Tilicho Lake's score is lower than the above-mentioned eight glacial lakes. For clarification, we have deleted the Tilicho Lake in the revised manuscript.

[redacted]

Google Earth image: Tilicho Lake

International Centre for Integrated Mountain Development (ICIMOD). Glacial Lakes and Glacial Lake Outburst Floods in Nepal; ICIMOD: Kathmandu, Nepal, 2011.

Khadka, N., Chen, X., Nie, Y., Thakuri, S., Zheng, G., & Zhang, G. Evaluation of Glacial Lake Outburst Flood Susceptibility Using Multi-Criteria Assessment Framework in Mahalangur Himalaya. *Front. in Earth Sci.* **8**, 601288 (2021).

Rounce, D. R., Watson, C. S., & McKinney, D. C. Identification of Hazard and Risk for Glacial Lakes in the Nepal Himalaya Using Satellite Imagery from 2000–2015. *Remote Sens.* **9**, 654 (2017).

Rounce, D. R., McKinney, D. C., Lala, J. M., Byers, A. C., & Watson, C. S. A new remote hazard and risk assessment framework for glacial lakes in the Nepal Himalaya. *Hydrol Earth Sys Sci.* **20**, 3455–3475 (2016).

My third concern is about Figure 4a (lines 176-178). It is not clear information about the hydropower and other infrastructures in the figure. Need to re-check.

Reply: Many thanks for the comment. The exposure index shown in Figure 4a is a comprehensive factor that combines the downstream buildings, hydropower projects, farmland, roads, and bridges. These five elements are not suitable to show separately in the same figure, as this will make the figure messy and reduce the aesthetics of mapping. Instead, we chose to count these elements on different areas (Supplementary Tables 5, 6). In addition, the complete exposure dataset (including hydropower and other infrastructures) with geographical information has been shared and can be viewed at the following link: <https://doi.org/10.5281/zenodo.8369266>. Scientists who are interested in the data can be easily accessed.

Supplementary Table 5 | Inundation area, potential disaster intensity and exposure of GLOFs in the Third Pole aggregated into the GTN-G regions. The exposure was quantified by building, hydropower, farmland, road, and bridge.

	Inundation area (km ²)	Reginal potential disaster intensity	Number of glacial lakes with high outburst potential	Building	Hydropower	Farmland (km ²)	Road (km)	Bridge
Altai	74.1	0.042	21	69	0	0	29	10
Hissar Alay	40.3	0.479	15	429	0	1.4	66	100
Pamir	290.3	0.16	62	1430	0	8.1	255	120
West Tian	510.5	0.211	120	10144	3	10.9	554	308
East Tian	360.6	0.155	87	3048	5	24.1	393	264
West Kunlun	110.6	0.022	7	0	0	0	1	0
East Kunlun	250.7	0.009	23	13	0	0	40	3
Qilian Shan	216.9	0.018	25	20	0	0	53	10
Inner Tibet	1513.4	0.08	164	1612	1	18.1	489	216
SE Tibet	470.7	0.383	123	10042	1	28.1	775	621
Hindu Kush	125.8	0.516	38	2858	7	16.7	170	329
Karakoram	210.2	0.193	64	1214	0	5.9	90	107
Western Himalaya	638.2	0.209	162	4555	19	17.4	424	425
Central Himalaya	374	0.203	126	3659	28	6	349	282
Eastern Himalaya	850.1	0.253	355	12671	40	49.3	942	914
Henduan Shan	316.4	0.237	107	4044	1	8	375	329

Supplementary Table 6 | Inundation area, potential disaster intensity and exposure of GLOFs in the Third Pole aggregated into the national areas. The exposure was quantified by building, hydropower, farmland, road, and bridge.

	Inundation area (km ²)	Reginal potential disaster intensity	Number of glacial lakes with high outburst potential	Building	Hydropower	Farmland (km ²)	Road (km)	Bridge
China	4080.3	0.191	880	25836	13	124.3	2940	2127
Bhutan	349.8	0.149	70	2738	3	5.1	211	90
Nepal	422.4	0.231	121	5840	43	7.9	303	385
India	422.4	0.239	164	4039	33	12	427	366
Pakistan	199.2	0.489	60	4274	8	20	187	407
Afghanistan	131	0.405	31	1514	2	7.6	167	165
Tajikistan	201.2	0.227	51	1417	0	7.3	210	158
Kyrgyzstan	387.9	0.239	74	955	0	7	293	172
Kazakhstan	101.2	0.207	32	9126	3	2.8	252	158
Mongolia	40.6	0.043	10	47	0	0	15	9
Russia	17.8	0.067	6	22	0	0	12	1

Reviewer #2 (Remarks to the Author):

This research offers an exhaustive review of glacial lake transformations, GLOF attributes, and risk evaluation in the Third Pole. In essence, the authors created a detailed catalogue of glacial lakes and assembled an enhanced GLOF dataset to understand their evolutions and interconnections. By employing numerical simulations and building a comprehensive exposure dataset, they methodically evaluated the dangers, vulnerability, and potential threats posed by glacial lakes in the Third Pole. Despite the great efforts from the authors, who compiled large amounts of valuable data for the region and run numerous simulations in each phase of their analysis, I detected several major issues, primarily associated with their risk analysis datasets and methodology, that need to be addressed before this manuscript could be accepted for publication.

Reply: Thank you for acknowledging the significance of our study. By considering both the general and specific comments provided by the three reviewers, we have undertaken substantial revisions of this manuscript. Most of our revisions focus on the Method section. This includes a detailed description of the HEC-RAS method, an explanation of the selection standards of the evaluation indicators, and a discussion of the uncertainties of the assessments and simulations.

The point-to-point responses to the reviewers' comments are provided below. The reviewers' comments and questions are in black font, and our responses are in blue font.

Major issues:

A) Is this a truly comprehensive assessment?

In the introduction (L41-45), the authors highlight a notable absence of a thorough dataset for glacial lakes in the specified region. They further suggest that existing knowledge might be influenced by biases and discrepancies, which I concur with. They also reference a study by Rick et al. (2022) (L358-362) that categorizes lakes into five distinct topological types. Given this context, one would anticipate a comprehensive susceptibility assessment encompassing all five lake types. This would ideally guide decision-makers in allocating resources based on the highest risk lakes. However, a subsequent section (L399-402) reveals that the authors chose to exclude two of these five types from their assessment. While I'm not necessarily advocating for the inclusion of these two types in their study, it's essential for the authors to provide clarity regarding which lake types their risk analysis refers to throughout their manuscript. Does it cover all the lakes in the region, or only a selective subset?

Reply: Thank you very much for the comment, which give us opportunity to clarify. In this study, a total of 5,894 glacial lakes were mapped in the Third Pole. These glacial lakes consist of five lake types: proglacial lakes, periglacial lakes, extraglacial lakes, supraglacial lakes, and ice-dammed lakes. Our risk assessment samples consist of 5,535 proglacial, periglacial, and extraglacial lakes, which were primarily dammed by a moraine or landslide. Supraglacial and ice-dammed lakes were not included in our evaluation scheme since they have different outburst mechanisms and are not among the main types in our inventory (N=361, only account for 6% of total number). Generally, their ice dams of these lakes are broken due to dam flotation or ice tunnel enlargement. This highlights the need for a separate evaluation system tailored to these lakes.

Moreover, GLOFs originating from ice-dammed lakes exhibit significant regional clustering and periodic drainage patterns in the Third Pole (Kingslake et al., 2017; Bazai et al., 2021). By directing our attention to specific basins or ice-dammed lakes with well-defined issues, such as the Kaygar, Shishper, and Merzbacher lakes

(Shangguan et al., 2017; Haemmig et al., 2017; Nie et al., 2023), we can effectively address their hazard and risk responses. Therefore, the pursuit of large-scale assessments of ice-dammed lakes appears to have comparatively limited utility compared to the more prevalent disasters and widely distributed hazard sources associated with moraine-dammed lakes in the Third Pole.

Bazai, N.A. et al. Increasing glacial lake outburst flood hazard in response to surge glaciers in the Karakoram. *Earth Sci. Rev.* **212**, 103432 (2021).

Haemmig, C. et al. Hazard assessment of glacial lake outburst floods from Kyagar glacier, Karakoram mountains, China. *Ann. Glaciol.* **55**, 34–44 (2017).

Kingslake, J., & Ng, F. Quantifying the predictability of the timing of jökulhlaups from Merzbacher Lake, Kyrgyzstan. *J. Glaciol.* **59**, 805–818 (2017).

Nie, Y. et al. Glacial lake outburst floods threaten Asia's infrastructure. *Sci. Bull.* **68**, 1361-1365 (2023).

Shangguan, D. H. et al. Quick release of internal water storage in a glacier leads to underestimation of the hazard potential of glacial lake outburst floods from lake Merzbacher in central Tian Shan Mountains. *Geophys. Res. Lett.* **44**, 9786-9795 (2017).

B) Decision analysis theory and rules for criteria

The integration of operations research methods like AHP, FCM, and TOPSIS in geosciences is expanding, offering valuable insights into risk assessments and intricate decision-making. However, with these techniques originating from distinct scientific disciplines, geoscientists must apply them judiciously to avoid misconceptions and unsound practices. A key point to understand is that the AHP is primarily suited for ranking and choice problems, not for sorting (Ishizaka et al., 2012). This limitation implies that while AHP can rank the relative risks among alternatives (e.g., glacial lakes), it won't objectively determine the inherent risk of an individual lake. This distinction is crucial; otherwise, resources might be misallocated based on perceived risks (as evidenced in Kougkoulos et al., 2018). When assessing GLOF susceptibility, the authors employ the Zhang et al. (2021) method that utilizes an FCM for weighting hazard criteria. While an intriguing approach, the rationale behind selecting FCM over other multi-criteria methods isn't clear. It would be beneficial to consult comprehensive works like Watrobski et al. (2018), which guide the selection of MCDA methods and could potentially enhance the robustness of their analysis. The same recommendation applies to the AHP technique employed later in the paper.

Reply: Thank you for your valuable comments and for recommending the references on MCDA methods. We have thoroughly examined the proposed references on the framework for MCDA method selection (Watrobski et al. 2018) and the rapid identification of hazardous glacial lakes with MCDA method (Kougkoulos et al., 2018), from which we benefit a lot in revising manuscript.

The methodology for assessing the GLOF susceptibility has been fully developed in recent years, achieving a transition from localized to large-scale, objective, and automated evaluation. The key components of the assessment process include the selection of efficient indicators, weighting schemes, classification methods, reliability validation, and sensitivity analysis. The overall methodology utilized in this study for assessing glacial lake hazards and risks is built upon the framework proposed by Allen et al. (2019) and subsequently refined by Zheng et al. (2021) and Zhang

et al. (2021). By reviewing over 50 different evaluation indicators in GLOF susceptibility assessment, we first identified 6 indicators with the strict rules of criteria selection inherent to MCDA, that is unbiased, exhaustiveness, non-redundancy, and consistency (same with Kougkoulos et al., 2018). These 6 indicators can be classified into five subsets based on the characteristics of the parent glacier, lake watershed, moraine dam, surrounding topography, and the glacial lake itself, providing a comprehensive depiction of the GLOF susceptibility.

Then, we assigned a weight to each indicator. It's important to clarify that the AHP and FCM methodology employed are solely utilized for calculating the relative weights of different indicators and are not involved in any glacial lake hazard sorting process. We chose AHP method in this study as they have been used in identifying potentially dangerous glacial lakes in the Third Pole region. The weight values of indicators calculated by AHP method are similar with the relevant percentage of the various GLOF triggers we counted. Furthermore, we also conducted a sensitivity analysis of the weighting schemes using different methods, e.g., AHP, FCM, and equal-weighting scheme, to verify the results. We show that assigning weights can increase the quality of the hazard level of drained glacial lakes, thus demonstrating a better GLOF susceptibility evaluation. Glacial lakes at high- and very high-hazard levels were not as sensitive to changes in different weighting schemes compared with those at medium- and low-hazard levels. In other words, weighting schemes have no decisive effect on the classification of high-hazard lakes (Zhang et al., 2021). We have discussed this in the revised manuscript.

Allen, S. K., Zhang, G., Wang, W., Yao, T. & Bolch, T. Potentially dangerous glacial lakes across the Tibetan Plateau revealed using a large-scale automated assessment approach. *Sci. Bull.* **64**, 435–445 (2019).

Kougkoulos, I. et al. Use of multi-criteria decision analysis to identify potentially dangerous glacial lakes. *Sci. Total Environ.* **621**, 1453–1466 (2018).

Wątróbski, J., Jankowski, J., Ziemia, P., Karczmarczyk, A., & Ziolo, M.. Generalised framework for multi-criteria method selection. *Omega* **86**, 107-124 (2019).

Zhang, T., Wang, W., Gao, T., An, B. & Yao, T. An integrative method for identifying potentially dangerous glacial lakes in the Himalayas. *Sci. Total Environ.* **806**, 150442 (2021).

Zheng, G. et al. Increasing risk of glacial lake outburst floods from future Third Pole deglaciation. *Nat. Clim. Change* **11**, 411–417 (2021).

Additionally, there's a concern with the selection of criterion (2) at L416-417. For an MCDA to remain unbiased, three primary properties must be maintained: a) exhaustivity, b) cohesion, and c) non-redundancy. The current criterion (2) appears to fall short due to concealed sub-criteria. Briefly, the risk of mass movement impacting a lake is influenced by a) the surrounding slopes (as pointed out by Zhang et al. regarding non-glacierized areas with a slope > 30°) but also by b) the proximity of these slopes to the lake. Consider two lakes: one surrounded by extensive steep terrain 500m away and another with a smaller steep area just 20m away. The modellers choice to score based solely on a) or to combine a) and b) together in a single criterion can skew results. To address this, the authors could introduce an additional criterion estimating the distance to non-glaciated slopes, especially since they've already incorporated a criterion for glacier-to-lake distance. I'd advise the authors to delve deeper into MCDA criteria selection theories, referencing key works like Hites et al. (2005), to enhance the robustness of their analysis.

Reply: Thank you for the comment. The criteria for indicator selection in this study is almost identical with Kougkoulos et al. (2018), that are exhaustiveness, non-redundancy and consistency. In this study, we adopted and

expanded upon the quantitative evaluation framework developed by Zhang et al. (2021). We initially identified 17 suitable indicators by setting preliminary criteria, such as data availability and accuracy, and non-redundancy. Subsequently, through an optimality analysis tailored to the Himalayan GLOF characteristics, we determined the best combination of indicators, including (1) the mean slope of the parent glacier, (2) the potential for a mass movement to strike a lake (for example, see Allen et al., 2019), excluding the glacier-influenced potential, (3) the mean slope of the moraine dam, (4) the watershed area, and (5) the lake perimeter. These indicators characterize a range of GLOF triggers, including ice avalanches, landslides or rockfalls, dam settlement or piping, heavy precipitation or upstream incoming water, and hydrostatic pressure. Considering the significant role of ice avalanches in triggering GLOFs and the decreased susceptibility of some glacial lakes due to glacier retreat in recent years, we added a new indicator, namely, the (6) horizontal distance between the glacier terminus and lake, to further highlight the impacts of glaciers. This indicator has been widely used and is highly effective in susceptibility assessment. The indicators used to assess the glacial lake susceptibility were carefully selected to correspond to various GLOF triggers, and the weights were assigned based on their importance, contributing to a more scientifically grounded assessment.

Regarding indicator (2) the potential for a mass movement to strike a lake, we employed the definition of Allen et al. (2019) that an impact into a lake is possible from any slope $>30^\circ$, where the overall slope trajectory is $>14^\circ$. Hence, this indicator inherently encompasses the information about the distance.

Allen, S. K., Zhang, G., Wang, W., Yao, T. & Bolch, T. Potentially dangerous glacial lakes across the Tibetan Plateau revealed using a large-scale automated assessment approach. *Sci. Bull.* **64**, 435–445 (2019).

Koukoulou, I. et al. Use of multi-criteria decision analysis to identify potentially dangerous glacial lakes. *Sci. Total Environ.* **621**, 1453–1466 (2018).

Zhang, T., Wang, W., Gao, T., An, B. & Yao, T. An integrative method for identifying potentially dangerous glacial lakes in the Himalayas. *Sci. Total Environ.* **806**, 150442 (2021).

Lastly, in L495-496 the authors use AHP and assign weights to the exposed infrastructure based on their importance. Could the authors please clarify how did they rank the importance? Is it based on expert judgment, personal judgment, or some actual data? At the moment it is not clear and this could change the final result, that is why I would appreciate if clarifications were given.

Reply: Thank you for your concern. We normalized each indicator to a range of 0–1 and employed the AHP method to allocate weights to the exposed buildings, hydropower projects, farmland, roads, and bridges. This ranking is based on expert judgement, which considers their varying levels of impacts on humans. Therefore, we list the importance of exposed infrastructure as follow: buildings, hydropower projects, farmland, roads, and bridges. By integrating these indicators, we derived the exposure index for glacial lakes. The GLOF risk is the result of multiplying their normalized hazard by the exposure. Both the exposure and risk were also classified into five categories: very low, low, medium, high, and very high, utilizing the natural Jenks approach.

C) GLOF simulation and hazard mapping clarifications

This section is impressive. The fact that the authors simulated 1500 GLOF propagation simulations in HEC-RAS shows true determination for a comprehensive assessment. Therefore, I salute this effort. Nevertheless, some points are not very clear to me. I know that this remains a large-scale assessment and detail is sometimes difficult (running multiple scenarios for each outburst would prove to be a very laborious task), but there is no clear indication of the

values used in the simulations. In L451-454 the authors indicate the shape of the hydrograph, but how was the Manning's value calculated? How uncertain are the flood results since no sensitivity analysis was conducted (based on multiple flood scenarios)? What about flood duration? Also, I would appreciate more technical details on the simulation time, the hardware used to run the simulations etc. All this would fulfil the criterion of reproducibility which is very important in our science and would pave the road for future studies.

Reply: We thank the reviewer for the positive comments here and the helpful recommendations. The HEC-RAS two-dimensional hydraulic model was selected to simulate the potential flood propagation. This model has successfully been used to reconstruct and evaluate many GLOF events worldwide, demonstrating its practicality and efficiency. By utilizing the two-dimensional flow surface, as well as a dam-breach hydrograph and a high-resolution digital elevation model (DEM), an unsteady flow simulation was conducted for each glacial lake with a high outburst potential.

For drainage volume of glacial lake, we estimated with an empirical equation which is determined by previous GLOFs. To maximize the GLOF simulation within a reasonable range, we utilized the upper limit curve of the scattered data points of the drainage volume and glacial lake volume ($V_d = 2.01V_t^{0.65}$) to determine the potential maximum drainage volume of each glacial lake (Supplementary Fig. 9). Subsequently, based on the estimated drainage volume and peak discharge, we adopted the assumption of a linear increase/decrease in the flood discharge to generate a dam-breach hydrograph (Westoby et al., 2014). Previous case studies have demonstrated that this hypothetical approach does not significantly affect the results.

Regarding the DEM used, we developed a composited scheme using two available DEMs for the Third Pole. The High Mountain Asia (HMA) DEM (8 m) was given priority as it was generated from very high-resolution commercial satellite imagery. The HMA DEM provides detailed ground features but has a limited coverage. Therefore, in areas where the HMA DEM is incomplete or missing, we utilized the Advanced Land Observing Satellite (ALOS) Phased Array type L-band Synthetic Aperture Radar (PALSAR) DEM (12.5 m) either for mosaicking with the HMA DEM or for direct use. All the DEMs were subjected to depression filling and surface smoothing processes to warrant unobstructed water flow.

Given the need for extensive GLOF simulations in the topographically complex Third Pole region, the model requires a robust and efficient set of parameters that allow the simulated flow to adapt to narrow valleys such as those in the Himalayas, as well as relatively flat surfaces such as Inner Tibet. Through preliminary sensitivity analysis, we determined the crucial parameters of the model to be as follows: the minimum computational unit is 30 m, Manning's coefficient is 0.06, and the computation interval is 5 s.

Plot of drainage volume versus the glacial lake volume

We also added a section on simulation uncertainty in the revised manuscript (see Robustness of assessments and simulations in Methods section). Due to the nature of conducting a maximum potential GLOF simulation, accurately quantifying uncertainties is challenging. There are two key sources of uncertainty to consider. The first is the assignment of model parameters. For example, we selected a relatively smaller Manning’s value to enhance the water flow efficiency. A sensitivity analysis performed in the reconstruction of the 1981 Cirenmaco GLOF in the Poiqu River Basin indicated that variations in the Manning’s value had no decisive impact on the flood inundation area and mean water depth (Wang et al., 2018). This feature is consistent with our modeling requirements. Second, the simulation accuracy of GLOFs is primarily determined by the high-resolution DEM as it serves as the main input data. There are essential differences between the results obtained from the commercial imagery produced HMA DEM and the freely available PALSAR DEM. GLOF propagation is significantly hindered by extensive depression filling on the surface of the PALSAR DEM, leading to underestimation of the flood inundation area, arrival time, and downstream exposure. Out of the 1,499 GLOF propagation simulations, the HMA DEM was utilized in 52.6%, which guarantees the fundamental credibility of our results. The mosaic production was used in 30.7% of the simulations, and the ALOS PALSAR DEM was only used in 16.5% of the simulations. Regionally, the HMA DEM was less frequently applied in the Altai and Southeast Tibet regions, limiting the reliability of the results for these regions. Overall, it should be noted that our maximum potential GLOF simulation and exposure analysis may still underestimate the actual risk due to data limitations in various areas.

Wang, W. et al. Integrated hazard assessment of Cirenmaco glacial lake in Zhangzangbo valley, Central Himalayas. *Geomorphology* **306**, 292–305 (2018).

Westoby, M. J. et al. Modelling outburst floods from moraine-dammed glacial lakes. *Earth Sci. Rev.* **134**, 137–159 (2014).

Minor comments:

L10: Change "cryosphere" to "cryospheric".

Reply: Changed.

L21-23: Please rephrase since this sentence is quite long and confusing.

Reply: This sentence has been revised as follows: “This study directly responds to the need for local disaster prevention and mitigation strategies, highlighting the urgent requirement of reducing GLOF threats in the Third Pole and the importance of regional cooperation.”

L26: Highly vulnerable to what?

Reply: We changed to “environmental vulnerability”.

L33-35: These are not the only two reasons causing an outburst flood.

Reply: This sentence has been revised as follows: “When a glacial lake is impacted by external forces, such as snow/ice avalanche, landslide, or rockfall, or is destabilized by continuous melting of the underlying buried ice in the moraine dam, it can suddenly release a large volume of water.”

L39: Is there any article in English illustrating these GLOFs?

Reply: Yes, here we cited a new global GLOF database: Lützow, N., Veh, G. & Korup, O. A global database of historic glacier lake outburst floods. *Earth Sys. Sci. Data* **15**, 2983–3000 (2023).

L41-42: Consider citing here the large review study by Emmer et al. (2022) which grouped representatives from 17 different countries and identified and elaborated trends and challenges and proposed possible ways forward to navigate future GLOF research.

Reply: Thank you. We cited the study here.

L111-114: These numbers add up to 97%. What about the remaining 3%?

Reply: About 10% of GLOFs are triggered by the melting of buried ice in moraine dams. We corrected the previous misconception of 7%.

L121: What exactly is GLOF intensity? Is it GLOF magnitude? Impacts from GLOFs? Flow velocity? Could you please explain.

Reply: Done - It is similar to the prior term “GLOF activity”, which refers to the frequency.

L121-L122: Could this be redundant? What if it is just the glacial lake area in a region? The number of lakes would basically just increase the area covered in a region.

Reply: The correlation between regional GLOF activity and the abundance of glacial lakes, along with their area changes, can prompt us to further assess the potential of future GLOFs. This warrants a more in-depth investigation.

L137: ...and that found 67 were. Delete “that”.

Reply: Done

L176-190: I think it would be nice to see also an estimation of the number of people affected. Even if it is a rough

estimate.

Reply: Thank you for your suggestion. By utilizing regional population distribution data, we estimated that roughly 190,000 lives are directly exposed within the GLOF paths. (see also in Line 190-191 of revised manuscript)

L240-241: There is a terminology confusion between hazard and risk here. I think that early warning systems should be a priority in glacial lakes that pose the highest risk. If a lake is only very susceptible to burst but doesn't have vulnerable elements downstream this ranks lower as a priority.

Reply: Thank you. We have corrected this terminology.

L357-363: The authors chose the classification by Rick et al. (2022) instead of classifications based on dam materials. I like the idea, but I would like to also see an argument behind this initiative.

Reply: This classification method emphasizes the connections between lakes and glaciers, facilitating the analysis of glacial lake dynamics within the context of glacial changes, compared to categorizing them based on their dam materials. We also added a sketch map on this. See Supplementary Fig. 6.

L395: In Supplementary Table 3 (Reconstruction of drainage volume for the historical GLOFs after 1980), we can see the area before outburst (in km²) but not after the drainage. I think both should appear.

Reply: We added the area after the drainage in the Supplementary Table 3.

L490-491: I appreciate the effort of the authors to manually digitize features for the entire third pole. First, I would like to ask if it would be possible to add this data in supplementary material (e.g. in .shp or other format) which would largely benefit future studies in the region. Second, they claim to have used high accuracy Maxar Premium Imagery for this task. Nevertheless, in my knowledge this imagery is not open-access, and I don't see any grant ID in the data availability section. Could the authors provide more information on how they acquired this data. Also, it would be highly appreciated if they indicated why they used a 200m buffer.

Reply: The complete exposure dataset (in shapefile) for this manuscript has been shared and can be viewed at the following link: <https://doi.org/10.5281/zenodo.8369266>. Currently, the Maxar Premium Imagery does not open on the OSM platform anymore, but it did when we were conducting the structure extraction during 2021-2022.

See also in **Line 467-474** of revised manuscript. With the exception of Nepal, which had better coverage, the completeness in the rest of the Third Pole was less than roughly 5%. Hence, we manually supplemented the structure features for the entire Third Pole. Considering the uncertainty of the modeled floodplain boundaries, a buffer distance of 200 m around the potential inundation areas of the GLOFs was incorporated to extract the exposure elements from high-accuracy Maxar Premium Imagery. Finally, we identified a total of 520610 buildings, 1996 km² of farmland, 98630 km of roads, and 6035 bridges within these buffers.

Figures

Fig.1: Delete Westerlies and Indian Monsoon (it adds unnecessary detail to the map). For area change rate I would suggest a Green-Yellow-Orange-Red colouring.

Reply: We deleted Westerlies and Indian Monsoon in the figure. For area change rate, we redraw the figure to enhance the contrast of the colour.

Fig. 1 Glacial lake distribution and changes in the Third Pole. **a** Maps highlighting the glacial lake expansion rate between 2018 and 2022. The pie charts with different colors and sizes illustrate the number and types of glacial lakes based on the Global Terrestrial Network for Glaciers (GTN-G) regions. **b, c** The histograms present the frequency of the glacial lakes in terms of their size (**b**) and elevation (**c**). **d, e** The smoothed density distribution of the area change rate for all of the glacial lakes and lakes with detectable changes (error > change area) in 2018–2022 (**d**) and 1990–2018 (**e**).

Fig. 2: This map needs to be presented in another way. The circles and dates are impossible to distinguish.

Reply: We enhanced the colour contrast to show the different years of outburst. We can clearly see the recent outburst floods (show with red colour) are distributed in central Himalayas and southeast Tibetan Plateau.

Fig. 2 Distribution of GLOFs in the Third Pole. **a** Map depicting the spatial and temporal distributions of 145 GLOFs in the Third Pole since 1900. The sizes of the hollow circles represent the drainage volumes of the GLOFs, which were reconstructed based on observed or estimated values (Supplementary Tables 2, 3). The colors of the hollow circles show the outburst year of the GLOFs. **b** A relatively complete subset of GLOFs occurring during 1981–2020, which was used to analyze the GLOF frequency and trends. **c** The regional abundance and area changes of the glacial lakes, except for those in the Altai, Tianshan, and Qilian mountains, were normalized and summarized within 0.1° longitude intervals. The regional GLOF number was counted at 1° . The glacial lake number and expanded area factors were employed to depict the regional abundance and area changes of glacial lakes, respectively. Their values were normalized to a range of 0–1.

Summary

This manuscript presents a commendable glacial lake risk assessment for the Third Pole. The authors' cross-disciplinary approach, the extensive data amassed, and the significant risk scoring findings clearly reflect their meticulous efforts. However, there are areas for important refinement, particularly regarding the clarity on the

specific types of glacial lakes analysed, the choice of operational research methods like FCM and AHP, and the somewhat sparse detailing and protocol for the HEC-RAS simulations.

Please, do not hesitate to contact me in case of questions (ikougkoulos@acg.edu)

Kind regards

Ioannis Kougkoulos

Reply: Dear Dr. Kougkoulos, thank you very much for your valuable comments. These comments are very useful guiding us to prepare revised manuscript. We have made substantial revisions following your suggestions and comments.

References to be considered:

Emmer, A., Allen, S. K., Carey, M., Frey, H., Huggel, C., Korup, O., ... & Yde, J. C. (2022). Progress and challenges in glacial lake outburst flood research (2017–2021): a research community perspective. *Natural Hazards and Earth System Sciences*, 22(9), 3041-3061. <https://doi.org/10.5194/nhess-22-3041-2022>

Hites, R., De Smet, Y., Risse, N., Salazar-Neumann, M., & Vincke, P. (2006). About the applicability of MCDA to some robustness problems. *European Journal of Operational Research*, 174(1), 322-332. <https://doi.org/10.1016/j.ejor.2005.01.031>

Ishizaka, A., Pearman, C., & Nemery, P. (2012). AHPSort: an AHP-based method for sorting problems. *International Journal of Production Research*, 50(17), 4767-4784. <https://doi.org/10.1080/00207543.2012.657966>

Wątróbski, J., Jankowski, J., Ziemia, P., Karczmarczyk, A., & Ziolo, M. (2019). Generalised framework for multi-criteria method selection. *Omega*, 86, 107-124. <https://doi.org/10.1016/j.omega.2018.07.004>

Reply: The references are properly cited in the revised manuscript.

Reviewer #3 (Remarks to the Author):

General comments:

In recent years, glacial lakes have become a research hot topic in the field of Earth Science due to their rapid expansion on a global scale and the resulting many significant GLOF disaster events. Understanding the current glacial lake risks is important for predicting the future risks, and it is of great importance to evaluate the increasing GLOFs hazard threats worldwide.

This study systematically sorted out the characteristics of current glacial lake activity (including changes and GLOF characteristics) in the Third Pole region, and quantitatively assessed the risk of downstream exposure elements caused by potential GLOFs. Based on a combination of high-precision modeling and detailed downstream exposure data, a comprehensive assessment was conducted to evaluate the GLOF hazards, exposure, and risks by employing an almost case-by-case methodology in this manuscript. It is a widely focused scientific topic with potentially wide audience, and I am particularly persuaded that the scientific findings supported by largely amount data, especially for integrated hazard evaluation method and the unprecedented exposure data of GLOFs hazards. In general, the assessment method and concept of this study are novel, and the conclusions drawn are also reliable. So, overall good achievements, such as comprehensive investigation of GLOF risks results, methodology for potential application for global glacial lake hazards evaluation outside the Third Pole region, several yielded valuable datasets, make me recommend this manuscript for a potential publication in Nature Communications after necessary revisions.

Reply: Thank you for acknowledging the significance of our study. By considering both the general and specific comments provided by the three reviewers, we have undertaken a revision of this manuscript. This includes a discussion of the HEC-RAS method, an explanation of the selection standards of the evaluation indicators, and a discussion of the uncertainties of the assessments and simulations.

The point-to-point responses to the reviewers' comments are provided below. The reviewers' comments and questions are in black font, and our responses are in blue font.

However, the structure of the first part of the method needs some adjustments. After describing the reasons for the current differences in the list of each dataset, it is necessary to explain the extraction standard of the glacial lake in this study, and then elaborate how to extract the glacial lake from the image. A comparison needs to be added to explain the advantages of the definition of a glacial lake (excellent by a sketch map to illustrate the difference of definitions and connections to glaciers). Furthermore, I think the uncertainty appraisals and discussions of the calculated lake volume and simulated GLOFs (depth, velocity, arrival time, etc.) are incomplete or even missing. I strongly suggest that discussions of uncertainties should be strengthened. Additionally, the authors set the minimum glacial lake threshold at 0.02 km² and excluded lakes within the glacier's 10 km buffer zone that were not hydraulically connected to the glacier, which made the dataset compact and suitable for evaluation. The authors should make comparison with previous studies to illustrate the differences and advantages between the glacial lake dataset in this study and the previous datasets. Multi sources and types of data with different resolutions are used from network, and some of them were examined or edited by the authors. The authors are also required to comprehensively describe their practicability, reliability and accuracy, and their possible or potential impacts on the results of GLOFs susceptibility, simulation, hazards mapping, exposure and risk.

Reply: Thank you for your valuable comments. We followed reviewer's suggestion to adjust the structure of the first part of the method. The modified version is as follows (see also in "Glacial lake mapping and change analysis" in

Methods).

(Line 282-300) Significant variations exist in the number of glacial lakes among the previously published inventories at the Third Pole scale. These differences can be attributed to varying area thresholds of 0.05–0.003 km², as well as different definitions of glacial lakes. Initially, glacial lakes were identified as those within a 10 km buffer of a glacier and with a hydraulic connection to a glacier. However, recent studies have included all lakes within the buffer regardless of glacier connections, resulting in inflated inventories. In this study, we needed to not only integrate previous research to analyze the state of the glacial lakes in the Third Pole but also to conduct a comprehensive and detailed risk assessment. We focused on glacial lakes with areas ≥ 0.02 km² that were primarily fed by contemporary glacier meltwater within a 10 km glacier buffer. Numerous thermokarst lakes and lakes without parent glaciers were excluded. Three time windows, 2018, 2020, and 2022, were selected to create new inventories and were combined with other available datasets to reveal the short-term and long-term glacial lake changes. A total of 878 Sentinel-2A/B images (10-m resolution) were used to manually delineate the glacial lakes (Supplementary Fig. 5). To ensure a sufficient storage period for the glacial lakes during the year and to minimize the presence of mountain shadows, priority was given to images captured between July and November with less than 10% cloud coverage. The images were processed using a false-color composition of bands 4, 3, and 2 to highlight the water bodies. The initial locations for the lake extraction were based on the 2018 glacial lake inventory created by Wang et al. (2020). Throughout the workflow, each glacial lake underwent thorough review at least six times, ensuring the inventory's completeness according to our standards.

(Line 331-335) Overall, our relatively compact inventory can illustrate the underestimated expansion of the glacial lakes (with glacier connections), compared to previous inventories. For instance, Wang et al. (2020) extracted data on all lakes within 10 km glacier buffers (with and without glacier connections) and reported an average increase of 15.2% in glacial lake area during the period 1990–2018 in the Third Pole, whereas our inventory reveals an increase of 30.5%.

We have also undertaken uncertainty analysis of assessments and simulations (see also in “Robustness of assessments and simulations” in Methods).

(Line 485-511) The indicators used to assess the glacial lake susceptibility have been carefully selected to correspond to various GLOF triggers, and weights have been assigned based on their importance. This contributes to a more scientifically grounded assessment. In the GLOF simulation scheme, glacial lakes without a high outburst potential were excluded. Previous studies have verified that over 90% of drained lakes identified as having a high or very high susceptibility. Moreover, in comparisons between different assessment cases, glacial lakes with a high outburst potential can be robustly identified even using different combinations of evaluation indicators and weighting schemes. These phenomena indicate that scaling down the sample for further GLOF simulations would not significantly impact the results as it covers all the interesting glacial lakes and ensures the accurate determination of future GLOF sources. For example, our simulation samples include all eight important glacial lakes in Nepal as identified Rounce et al. (2016), and our assessment of the GLOF threat to Almaty City agrees with Bloch et al. (2011).

Due to the nature of conducting a maximum potential GLOF simulation, quantifying uncertainties is challenging. There are two key sources of uncertainty to consider. The first is the assignment of model parameters. For example, we selected a relatively smaller Manning's value to enhance the water flow efficiency, with less emphasis on its physical significance. A sensitivity analysis performed in the reconstruction of the 1981 Cirenmaco GLOF in the Poiqu River Basin indicated that variations in the Manning's value had no decisive impact on the flood inundation area and mean water depth (Wang et al., 2018). This conclusion aligns with our modeling requirements. Second, the simulation accuracy of GLOFs is primarily determined by the high-resolution DEM as it serves as the main input

data. There are essential differences between the results obtained from the commercial imagery produced HMA DEM and freely available PALSAR DEM. GLOF propagation is significantly hindered by extensive depression filling on the surface of the PALSAR DEM, leading to underestimations of the flood inundation area, arrival time, and downstream exposure. Out of the 1,499 GLOF propagation simulations, the HMA DEM was utilized of 52.6%, which guarantees the fundamental credibility of our results. The mosaic production was used in 30.7% of the simulations, and the ALOS PALSAR DEM was only used in 16.5% of the simulations. Regionally, the HMA DEM was less frequently applied in the Altai and Southeast Tibet regions, limiting the reliability of their results. Overall, it should be noted that our maximum potential GLOF simulation and exposure analysis may still underestimate the actual risk due to data limitations in various areas.

Specific comments:

L65-69, Linguistic streamlining without too many repetitive sentences. I suggested providing a sketch map (extended figure) as extended Figure to illustrate the definitions and differences of five types of glacial lakes, including their connections to glaciers.

Reply: Thank you for the good suggestion. We asked a scientific editing service, Letpub (www.letpub.com), to polish the language of the manuscript. We also draw a sketch map to illustrate the different types of glacial lakes discussed in the manuscript. See also in Supplementary Fig. 6.

Supplementary Figure 6 The various types of glacial lakes. Based on their topological positions relative to their parent glaciers, the glacial lakes are classified into five types, namely, proglacial, periglacial, extraglacial, supraglacial, and ice-dammed lakes.

L100-101, more information about the relationships of moraine-dammed lake, proglacial lake and periglacial lake is needed. Here you only focused on moraine-dammed lakes or the proglacial and periglacial lakes that were moraine-dammed lakes?

Reply: Thank you for the concern. The proglacial and periglacial lakes are generally dammed by moraines in the Third Pole. Here we focused on proglacial and periglacial lakes that were dammed by moraine.

L104-105, Is there likelihood that more GLOFs events are recorded because of the increasing capacity of satellite monitoring or other reasons? As it was also reported that no distinct GLOFs frequency was detected in Hindu Kush–Karakoram–Himalaya–Nyainqentanglha region (Veh et al., 2019).

Reply: Thank you for the comment. Compared to the study of Veh et al. (2019), our GLOF dataset increases the number of GLOFs from 39 to 73 in the Hindu Kush–Karakoram–Himalaya–Nyainqentanglha region after the 1980s. This progress is, of course, attributable to the increasing capacity of satellite monitoring and the increased research interest. In this regard, our detected increased frequency of GLOFs after the 1980s is credible in this region.

L 115-116, What types these short-lived glacial lakes are? They are moraine-dammed lake? Why there exist ice-tunnels? I did not carefully read the documents of 28-30, however, in my knowledge, the trigger of ice tunnels usually appeared in ice-dammed lakes.

Reply: Thank you for the comment. These short-lived glacial lakes are typically dammed by moraines and are categorized as proglacial or periglacial lakes. The primary trigger for GLOFs in this region is the movement of ice tunnels, mainly because these moraine dams contain buried ice. It should be noted that the GLOFs in the Tien Shan Mountains differ from those in the Himalayas and have their own unique characteristics. Due to their relatively lower magnitude, we have had limited opportunities to validate the geomorphic GLOF diagnostic features using satellite images, such as breached dams, outwash fans, and downstream impact areas.

L121, The regional abundance of glacial lakes can be expressed using specific factors, such lake number, area, and volume, etc., please point out which factor used in figure 2c. The area change rate is in the same situation.

Reply: Thank you for the question. The glacial lake number and expanded area factors were employed to depict the regional abundance and area changes of the glacial lakes, respectively. Their values were normalized to a range of 0–1.

L 130-132, in Figure 2c, the unit of y-axis is indistinct. Are they in frequency or percentage?

Reply: The regional abundance and area changes of glacial lakes were normalized to a range of 0–1 and summarized within 0.1° longitude intervals.

L168, Please note that these 1499 glacial lakes belong to high or very high susceptibility.

Reply: Revised.

L182, Add latitude and longitude to the first occurrence of a place or glacial lake name.

Reply: Done.

L 193-194, what the bars indicate in Fig.4b and c?

Reply: Fig. 4b is the box plot of the distance between each glacial lake and the nearest human community downstream that is exposed, aggregated by region. Fig. 4c is the box plot of the early warning time, aggregated by region. Bars indicate the maximum and minimum of the data.

L204-211, Logical order may be problematic.

Reply: The modified sentences are as follows:

(see also in **Line 222-227** of revised manuscript) Currently, the number and area of the contemporary glacial lakes in these regions are smaller, and the GLOF frequency and magnitude are much lower compared to those in the Eastern Himalayas and Southeastern Tibet. However, when the dramatic potential for the development of glacial lakes in the western Third Pole under future climate change scenarios is considered, the trend of the hazard and risk extension from east to west will be supported by the abundance of glacial lakes and downstream exposure.

L 344, How do you determine and practically operate connections of glaciers and lakes?

Reply: We utilize Sentinel-2 images acquired in 2022 to manually determine and classify the connections between the glaciers and lakes.

L358, Note the complete use of brackets.

Reply: Corrected.

L 366-367, How about the uncertainty of the calculated lake volume?

Reply: The glacial lake volume was estimated using a set of empirical equations proposed by Zhang et al. (2023), and its uncertainty was calculated by using the 95% confidence intervals of the regression curves.

L 417, Maybe, the indicator of mean downstream slope of the moraine dam is more pertinent than the mean slope of the moraine dam.

Reply: Yes, both indicators are highly efficient in reflecting the dam stability. The indicator of the mean slope of the moraine dam has been validated in the Himalayan region and delineated easily in the remote sensing image. So, we use mean slope of the moraine dam in this study.

L 419-420, the indicator of horizontal distance between the glacier terminus and lake is not newly used. As far as my knowledge, at least it is used in Wang et al., 2008, 2012, 2013 where 0.5km and 1km were used as thresholds.

Reply: Thank you for the comment. We have added the related references.

L486, Please add the weblink of OpenStreetMap.

Reply: Done.

L535, Some papers may be finished their discussion in AGU and accepted now, please write the complete reference information.

Reply: Thank you for the information. We have included the complete reference information.

L603, Uniformity of color and size of symbols in this figure.

Reply: Done. See also in Supplementary Fig. 5.

Supplementary Figure 5 Overview of Landsat and Sentinel images used for mapping the glacial lakes. (a) A total of 878 Sentinel-2A/B images covering all of the glaciated regions in the Third Pole were used to delineate the glacial lakes in 2018, 2020, and 2022; and (b) 313 individual scenes from Landsat missions 4, 5, 7, and 8 were used to identify glacial lakes overlooked in other available glacial lake datasets. (c) Temporal distribution of these images.

REVIEWERS' COMMENTS

Reviewer #1 (Remarks to the Author):

The author's efforts in the study of glacial lakes and their outburst flood in the Third Pole are very commendable. As mentioned in the manuscript it is a comprehensive study which was lacking in the past in such a holistic approach. I am satisfied with the updates to the manuscript and recommend publishing it.

Reviewer #2 (Remarks to the Author):

I wish to express my appreciation to the authors for their diligent efforts in refining the paper within such a short timeframe. It's commendable to note that the authors have adeptly incorporated many of my suggestions. Through the constructive feedback from reviewers and the dedicated work of the authors, the manuscript has seen significant improvement.

Regarding the major issues I previously outlined, namely A and C, I am satisfied with the depth and seriousness with which they have been addressed and have no further comments on these points. On major issue B, there remains a broader concern that should be taken seriously in future research efforts. The prevalent use of AHP methodologies in Earth Science, owing to their accessible approach, requires a closer examination. In my opinion, the GLOF research community should endeavour to adopt more tailored methods and seek consultation from experts in operations research before applying such techniques. This would ensure the employment of the most current and appropriate methodology.

Also, I wish to express my gratitude to the authors for their thorough attention to the minor issues I highlighted. Their commitment to addressing every concern is praiseworthy.

I'd like to reiterate my earlier sentiments: this manuscript represents an impressive endeavour from a risk analysis viewpoint. It not only introduces a fundamental theoretical advancement but also promises tangible operational benefits for communities facing threats from GLOFs and recent glacier change.

Based on the aforementioned considerations, I recommend the paper be accepted. However, I would advise a comprehensive review of the English language, as there are occasional syntactical and spelling errors present throughout the manuscript.

Please, do not hesitate to contact me in case of questions (ikougkoulos@acg.edu).

Kind regards,
Ioannis Kougkoulos

Reviewer #3 (Remarks to the Author):

I have carefully read the revised manuscript and the responses document to reviews. I think the manuscript has been substantially improved in discussion of the uncertainties of the assessments and simulations and the Robustness of the method of GLOF susceptibility. All my comments were appropriately responded. So, I recommend this updated manuscript as a potential publication in Nature Communications after minor revision.

L75-76, more detailed information (e.g., percentage) is suggested to added.

L161, is the probability of 0.24 somewhat a threshold value?

L270, what is the number of reference?

L350, is there any bed rock dam among the 5535 lakes?

Reviewer #1 (Remarks to the Author):

The author's efforts in the study of glacial lakes and their outburst flood in the Third Pole are very commendable. As mentioned in the manuscript it is a comprehensive study which was lacking in the past in such a holistic approach. I am satisfied with the updates to the manuscript and recommend publishing it.

Reply: Thank you for acknowledging the significance of our study and supporting its publication.

Reviewer #2 (Remarks to the Author):

I wish to express my appreciation to the authors for their diligent efforts in refining the paper within such a short timeframe. It's commendable to note that the authors have adeptly incorporated many of my suggestions. Through the constructive feedback from reviewers and the dedicated work of the authors, the manuscript has seen significant improvement. Regarding the major issues I previously outlined, namely A and C, I am satisfied with the depth and seriousness with which they have been addressed and have no further comments on these points. On major issue B, there remains a broader concern that should be taken seriously in future research efforts. The prevalent use of AHP methodologies in Earth Science, owing to their accessible approach, requires a closer examination. In my opinion, the GLOF research community should endeavour to adopt more tailored methods and seek consultation from experts in operations research before applying such techniques. This would ensure the employment of the most current and appropriate methodology.

Reply: Thank you for your valuable comments. We fully agree with your suggestions on how to use the AHP and have emphasized this in the manuscript as well. See Section of GLOF susceptibility, *“While like AHP and some other multi-criteria decision analysis methodologies are prevalent in Earth Science, for relatively complex assessment cases, it should be applied cautiously, with input from experts in operations research to ensure the use of the most current and appropriate methodology.”*

Also, I wish to express my gratitude to the authors for their thorough attention to the minor issues I highlighted. Their commitment to addressing every concern is praiseworthy. I'd like to reiterate my earlier sentiments: this manuscript represents an impressive endeavour from a risk analysis viewpoint. It not only introduces a fundamental theoretical advancement but also promises tangible operational benefits for communities facing threats from GLOFs and recent glacier change. Based on the aforementioned considerations, I recommend the paper be accepted. However, I would advise a comprehensive review of the English language, as there are occasional syntactical and spelling errors present throughout the manuscript.

Reply: Thank you for acknowledging the significance of our study and supporting its publication. The English language of the manuscript has been previously touched up by professional organizations. To ensure the smoothness of the manuscript, we once again scrutinized the terminology and spelling of the words throughout the manuscript.

Please, do not hesitate to contact me in case of questions (ikougkoulos@acg.edu).

Kind regards,

Ioannis Kougkoulos

Reviewer #3 (Remarks to the Author):

I have carefully read the revised manuscript and the responses document to reviews. I think the manuscript has been substantially improved in discussion of the uncertainties of the assessments and simulations and the Robustness of the method of GLOF susceptibility. All my comments were appropriate responded. So, I recommend this updated manuscript as a potential publication in Nature Communications after minor revision.

Reply: Thank you for acknowledging the significance of our study and supporting its publication.

L75-76, more detailed information (e.g., percentage) is suggested to added.

Reply: We have incorporated the mean proportion value of extraglacial and periglacial lakes in most regions of the Tibetan Plateau.

L161, is the probability of 0.24 somewhat a threshold value?

Reply: This is just an example illustrating that a GLOF probability of 0.24 indicates a region with the potential for at least five GLOF impacts from different sources.

L270, what is the number of reference?

Reply: References 45 and 46 have been re-added.

L350, is there any bed rock dam among the 5535 lakes?

Reply: In the Tibetan Plateau, bedrock-dammed lakes are not the primary types of glacial lakes, and it's challenging to distinguish them from moraine-dammed lakes using satellite images. Therefore, throughout the manuscript, we do not take this into account.